# Investigation of the Solid-Phase Joint of VT-14 Titanium Alloy with 12KH18N10T Stainless Steel Obtained by Diffusion Welding through Intermediate Layers

Alexander Viktorovich Lavrishchev [1], Sergei Viktorovich Prokopev [2], Vadim Sergeevich Tynchenko [1,3,4,*], Aleksander Vladimirovich Myrugin [1], Vladislav Viktorovich Kukartsev [5,6], Kirill Aleksandrovich Bashmur [3], Roman Borisovich Sergienko [7], Valeriya Valerievna Tynchenko [5,8] and Aleksey Vasilyevich Lysyannikov [9]

1   Information-Control Systems Department, Institute of Computer Science and Telecommunications, Reshetnev Siberian State University of Science and Technology, 660037 Krasnoyarsk, Russia; lav@optilink.pro (A.V.L.); avm514@mail.ru (A.V.M.)
2   Welding of Aircraft Department, Institute of Mechanical Engineering and Mechatronics, Reshetnev Siberian State University of Science and Technology, 660037 Krasnoyarsk, Russia; prokopiev@sibsau.ru
3   Department of Technological Machines and Equipment of Oil and Gas Complex, School of Petroleum and Natural Gas Engineering, Siberian Federal University, 660041 Krasnoyarsk, Russia; bashmur@bk.ru
4   Digital Material Science: New Materials and Technologies, Bauman Moscow State Technical University, 105005 Moscow, Russia
5   Department of Informatics, Institute of Space and Information Technologies, Siberian Federal University, 660041 Krasnoyarsk, Russia; vlad_saa_2000@mail.ru (V.V.K.); 051301@mail.ru (V.V.T.)
6   Department of Information Economic Systems, Institute of Engineering and Economics, Reshetnev Siberian State University of Science and Technology, 660037 Krasnoyarsk, Russia
7   Machine Learning Department, Gini Gmbh, 80339 Munich, Germany; roman@gini.net
8   Department of Computer Science and Computer Engineering, Institute of Computer Science and Telecommunications, Reshetnev Siberian State University of Science and Technology, 660037 Krasnoyarsk, Russia
9   Department of Aviation Fuels and Lubricants, School of Petroleum and Natural Gas Engineering, Siberian Federal University, 660041 Krasnoyarsk, Russia; av.lysyannikov@mail.ru
*   Correspondence: vadimond@mail.ru; Tel.: +7-95-0973-0264

**Abstract:** This paper describes the technological process of manufacturing bimetallic billets, which are capable of operating at high pressures, high temperatures, and in corrosive environments, from VT-14 titanium alloy and 12KH18N10T stainless steel. To obtain a joint with a strength of at least 350 MPa, the diffusion welding method was used, which makes it possible to obtain equal-strength joints using dissimilar materials. The connection of VT-14 titanium alloy with 12KH18N10T stainless steel after obtaining bimetallic billets with the desired properties was investigated. We studied the welded VT-14 and 12KH18N10T joint obtained by diffusion welding through intermediate spacers of niobium Nb (NbStrip-1) and copper Cu (M1). On the basis of our investigations, the optimum welding modes are as follows: welding temperature: 1137 K; welding pressure: 18 MPa; welding time: 1200 s. Mechanical tests, tightness tests, and metallographic, factographic, and micro-X-ray structural studies were carried out, the results of which indicate the effectiveness of the proposed approach.

**Keywords:** diffusion welding; titanium alloy; stainless steel; solid-phase joint; intermediate layer; technology

## 1. Introduction

In atomic power, space technology using pipelines, and other products made of titanium alloys, there is an urgent need to connect titanium pipes with steel ones.

The high reactivity of titanium at high temperatures, the nature of its interaction with iron, including limited mutual solubility in the solid state, and the presence of a low-melting eutectic and several intermetallic compounds [1–3] create significant technological difficulties in the manufacture of bimetallic titanium-steel compounds by welding methods.

When obtaining brazed steel-titanium joints, there are several factors that negatively affect the manufacturability and the cost of their manufacture. These factors include the use of silver as a solder, as well as the use of threads of an increased accuracy class.

Direct diffusion welding of a titanium alloy with stainless steel leads to residual stresses in the solid-phase joint zone due to a mismatch in the thermal expansion coefficients of the materials being joined and the formation of brittle intermetallic phases in the diffusion zone [4–6]. Mutual diffusion between titanium and stainless steel is carried out by the migration of atoms of the same chemical type across the plane of the joint and causes the formation of intermetallic compounds based on Fe + Cr + Ti and Fe + Ti in the reaction zone; these brittle intermetallic compounds worsen the mechanical properties of the formed compound [1]. Therefore, the use of an intermediate material that prevents the formation of particularly brittle intermetallic phases in the process of diffusion welding is of decisive importance for the quality of the welded joint.

The advantages and disadvantages of titanium are well known. It is common to compensate for the disadvantages by creating composite materials, for example titanium alloys [7–9]. At present, solid-state welding methods are widely used, which allow one to control the processes occurring at the interface and the interface of dissimilar materials, and thus ensure a high-quality connection to a large extent [10–12].

The majority of existing studies exploring the formation of a welded joint during diffusion welding of dissimilar materials [2,3,13] highlight the following as being necessary: first, the presence of physical contact between the clean surfaces to be joined; second, control of the variation of these surfaces caused by plastic deformation of the contacting layers under creep conditions; third, the occurrence of diffusion processes and processes of physicochemical interaction at the interface and in the adjacent zones of dissimilar materials, leading to the formation of the structure of the welded joint.

Oxide films, adsorbed layers, and foreign contamination on the surfaces to be welded prevent the occurrence of physical contact [14–16]. When machining surfaces for welding (grinding, polishing, etc.), thick oxide films and adsorbed layers (in the case of titanium) do not generally interfere with welding, since the former quickly dissolve due to the high solubility of oxygen in titanium at welding temperatures, and the latter evaporate or are absorbed by the metal when heated. As a result of its high chemical affinity for oxygen, titanium can reduce oxide films on the surface of other metals in contact with it. To prevent saturation and oxidation of titanium and materials welded to it with gases, diffusion welding is carried out in a vacuum (0.13 Pa) or, in some cases, in an inert atmosphere [2,17,18].

The purpose of this work is to study the possibilities of joining VT-14 titanium alloy with 12KH18N10T stainless steel by diffusion welding and to develop a technological process for producing billets of bimetallic adapters (Figure 1) with specified properties by diffusion welding instead of the currently used necks made of titanium bimetallic plates. hot rolled steel under vacuum.

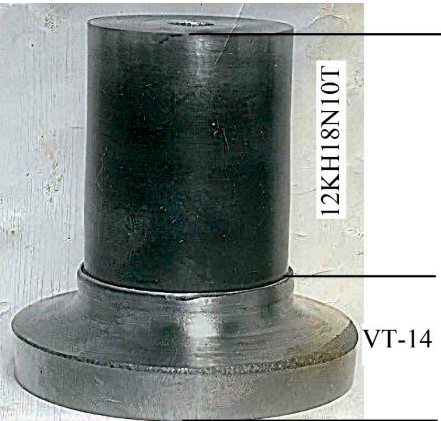

**Figure 1.** Welded bimetal adapter billet.

A necessary requirement for high-quality diffusion welding is the thorough preparation of the surfaces to be welded. In addition to degreasing and cleaning, a certain surface roughness should be achieved. It is generally accepted that the less rough the surfaces to be joined, the easier it is to obtain a high-quality connection through diffusion welding [19–21]. This is due to a decrease in the size and curvature of the formed microvoids and their more complete "healing" in the final stage of welding [2,22].

The bond strength of dissimilar materials is determined by the nature of changes in the composition, structure, and properties in the welding zone. If they change monotonically and continuously, which only occurs in cases of mutual unlimited solubility of the materials to be welded, then, with optimal welding parameters, the strength of the joint will not be lower than the strength of the weakest of the materials being welded. The bond strength is only ensured with very thin intermediate intermetallic layers, the thickness of which does not exceed the critical one. The thickness of the intermetallic layers, as shown in many studies [1,2,23], increases with increasing temperature and the duration of welding.

An effective way to prevent and form brittle layers is the use of intermediate spacers during diffusion welding. Similar gaskets are introduced in the form of foil and powders, which are deposited on or applied to the surfaces to be welded [2,24]. Often, such pads are designed as a "sandwich" consisting of several layers, each of which performs a different function.

To prevent intermetallic layers in diffusion welding of titanium and steel, various intermediate spacers in the form of foil are used, which are placed before welding. Copper foil is laid on the steel side, and vanadium (niobium, tantalum) foil on the titanium side. There are various disadvantages of this design. For example, it is impossible to achieve sufficient welding forces for plastic deformation of refractory foils (niobium, vanadium, tantalum) during traditional diffusion welding (5–15 MPa) without causing macrodeformation of the entire adapter. As a result, defects such as non-penetration, instability, and a decrease in the strength and tightness of welded joints occur. In addition, before stacking, foils must be subjected to joint vacuum rolling, which complicates and increases the cost of the process [2]. Moreover, the optimum ranges of thickness of intermediate foils, which ensure the absence of brittle intermetallic and eutectic layers, have not been determined.

The main difficulties in welding titanium and its alloys are related to the absorption of gases, their diffusion from the base metal, an increase in their content in the welding zone, and structural transformations. The high reactivity of titanium at high temperatures in relation to oxygen, nitrogen, and hydrogen reduces the ductility of the metal, causing cracks and brittle fractures [1–3].

The mechanical characteristics of butt joints formed by diffusion welding in a vacuum using VT1-0+12KH18N9T and OT4+12KH18N9T (temperature 1023–1123 K, welding time 900 s) are poorer than those of the base materials. The use of vanadium and copper gaskets in welding VT6 or VT5-1 with steel 12KH18N9T makes it possible to obtain an ultimate strength of up to 530–570 MPa. No intermetallic phases are found in the compound even after prolonged heating at high temperatures (1273 K for 10 h). During welding, the copper layer prevents the formation of vanadium carbides, which embrittle the joints. In the vanadium–copper compound, low-melting compounds and intermetallic compounds are not formed. To obtain stable results, it is advisable to use a thin multilayer tape (Ti + Cu + Ni), which is obtained by hot rolling in a vacuum, as a cushioning material. As a result of this tape, the tensile strength of VT5-1 and AT3 joints with 12KH18N10T steel in tension is 500–590 MPa [5].

Rolling welding is carried out in a vacuum. The negative influence of carbon on the mechanical characteristics of the joint is demonstrated by the formation of titanium carbide (TiC). An increase in the carbon content in steel from 0.02 to 0.45% leads to a decrease in the strength from 260 to 140 MPa. When using vanadium spacers, the carbon content must be <0.02% [5].

When welding VT6 with 12KH18N10T steel with a combined gasket made of Nb + Cu (vacuum: 0.00266 Pa; temperature: 623 K; degree of reduction: 45–50%), joints of equal

strength are obtained (destruction of samples during testing: in copper). On the join of niobium with titanium, zones of solid solutions are formed, which are characterized by an increased hardness. On the niobium and copper joint, there is a diffusion zone with a length of about 40 μm. In the niobium-titanium transition, a diffusion zone is not observed. The thickness of the niobium gaskets is on the order of 0.05–0.2 mm, and for copper, the thickness is 0.1–0.46 mm [2,3].

Joints with superior plasticity can be obtained by welding titanium with zirconium, niobium, and tantalum. In diffusion welding of titanium with steel, as in fusion welding, it is necessary to use intermediate layers, in order to avoid the formation of brittle layers in the contact zone. Niobium-copper, tantalum-copper, vanadium-copper, etc., can be used as interlayers. One of the methods of solving the problem of joining titanium to steel is the use of adapters made of bimetallic strips with a layer thickness ratio of 1:1 during welding.

Positive results can be obtained using pressure welding methods, as well as barrier layers and inserts from a third metal that does not form brittle phases with the materials being welded at high temperatures.

A double gasket made of vanadium or niobium on the titanium side and copper on the steel side was used. The compound does not show intermetallic phases even after prolonged heating at high temperatures. The copper layer during welding prevents the formation of vanadium carbides, which embrittle the joints. In the vanadium-copper compound, low-melting compounds and intermetallic compounds are not formed.

At the interface between niobium and titanium, zones of solid solutions with increased hardness are formed. In addition, on the border of niobium and copper, there is a diffusion zone with a length of about 40 microns. In the niobium-titanium transition, the diffusion zone is not observed. The thickness of the niobium gaskets is taken on the order of 0.2–0.5 mm, copper 0.1–0.3 mm.

The strength of the diffusion layer is higher than that of copper and steel. The presence of diffusion zones at the steel + copper and copper + niobium interface is indirectly confirmed by the results of tensile tests, which showed that the destruction of the samples in all cases occurs over the entire area of the samples along a less strong material-copper.

The technology for rolling these strips has been developed in various organizations [2,5]. Thus, the following materials were identified:

- Titanium alloy: VT-14 (Russian OST1 90013-81 [25]), where hydrogen content is no more than 0.01% (the closest materials are T-A4D3V in France and 4Al-3Mo-1V in the USA);
- Steel: 12KH18N10T (Russian State Standard 7350-77 [26]);
- Niobium gasket: NbStrip-1 (Russian TU 48-4-317-74 [27]);
- Copper gasket: M1 (Russian Interstate Standard 1173-2006 [28]).

## 2. Materials and Methods

To obtain stable weld quality, we adhered to the following technical requirements for the production of the 12KH18N10T+Cu+Nb+VT-14 bimetallic adapter by diffusion welding. The peel strength of the layers is shown in Table 1.

**Table 1.** Tensile strength characteristics of the joint at various temperatures.

| Test temperature (K) | 77 K | 173 K | 223 K | 253 K | 373 K | 473 K |
|---|---|---|---|---|---|---|
| Strength, not less than (MPa) | 640 | 540 | 450 | 300 | 270 | 250 |

Destruction along the boundary of the layers (interlayer of copper or niobium) was not permissible, as in this case, there was a poor-quality connection. The impact strength KCU was not less than 250 kJ/m$^2$. The peel strength of the layers under cyclic loading is shown in Table 2.

**Table 2.** Strength characteristics of the connection under cyclic loading.

| Tension (MPa) | Cycles to Failure |
|---|---|
| 400 | 0.39e03 |
| 300 | 11e03–30e03 |
| 250 | 10e03–62e03 |
| 200 | 34e03–231e03 |
| 150 | 228e03–1000e03 |

To ensure a high-quality product, the following requirements must be met:

1. The content of gas impurities in the titanium component of the adapter cannot exceed the following after welding:

   - For $O_2$: 0.15%;
   - For $H_2$: 0.01%;
   - For $N_2$: 0.05%;

2. The microstructure of the titanium component of the adapter corresponds to the 1–7 type of the nine typical scale (OST 97 9465-81 [29]);
3. The adapter material provides the required level of properties after heating to 673 K three times in the interlayer joint zone;
4. The connection of the adapter layers is continuous over the entire area;
5. The bimetallic adapter remains functional:

   - After cyclic loaded: 300 cycles;
   - $P_{rab.max}$ = 36 MPa.

VT-14 belongs to the third class. In terms of strength, VT-14 belongs to the class of high-strength alloys. By designation, VT-14 titanium alloy can be classified as a weldable structural alloy. Titanium billets were made from hot-rolled VT-14 bar (diameter 45 mm), the composition of which is indicated in Table 3.

**Table 3.** Chemical composition of VT-14 bar in wt. %.

| Ti | Al | Mo | V | C | Fe | Si | Zr | O | N | H |
|---|---|---|---|---|---|---|---|---|---|---|
| main | 5.7 | 3.3 | 1.5 | 0.10 | 0.7 | 0.05 | 0.06 | <0.15 | <0.05 | <0.02 |

The rod grades were checked with a stillscope according to the content of vanadium and aluminum, and the content of $H_2$ was determined to be 0.005%. The mechanical properties of VT-14 alloy are given in Table 4.

**Table 4.** Mechanical properties of VT-14 alloy.

| Mechanical Properties of VT-14 alloy | $\sigma_{ten}$ (MPa) | δ (%) |
|---|---|---|
| Annealed condition | 900–1070 | 8 |
| Tempered and aged | 1200 | 6 |

12KH18N10T stainless steel belongs to the austenitic class of stainless steels. The widespread use of austenitic stainless steels is a consequence of its impressive anticorrosion, mechanical, and technological properties.

Chromium improves hardening, and nickel increases the toughness [4]. An increase in the carbon content increases the strength of the steel, but decreases its toughness, thereby diminishing the effect of nickel. The chemical composition and physical and mechanical properties of steel are given in Tables 5 and 6.

**Table 5.** Chemical composition of 12KH18N10T stainless steel in %.

| Fe | Ni | Cr | C | Si | Mn | Ti | S | P |
|---|---|---|---|---|---|---|---|---|
| 62–66 | 9–11 | 17–19 | 0.12 | 0.8 | 2.0 | 0.7 | 0.02 | 0.035 |

**Table 6.** Physical and mechanical properties of 12KH18N10T steel.

| Indicator | Designation | The Quantity |
|---|---|---|
| Melting temperature (K) | $T_{mel}$ | 1873 |
| Specific gravity (kg/m$^3$) | $\gamma$ | 7920 |
| Scale resistance (K) | T | 1148 |
| Thermal expansion coefficient (K$^{-1}$) | $\alpha$ | 16.6e-6 |
| Thermal conductivity (W/m*K) | $\lambda$ | 15 |
| Normal elastic modulus (MPa) | E | 2.03e5 |
| Temperature range of deformations (K) | - | 1396–2346 |
| Tensile strength (MPa) | $\sigma_{ten}$ | 550 |
| Non-magnetic | - | - |

Niobium is oxidized at air temperatures above 473 K. It interacts with nitrogen at temperatures above 673 K. At 293 K, niobium absorbs up to 104 cm$^3$/g of hydrogen; at >1273 K, hydrogen essentially does not dissolve. Niobium forms carbide with carbon at high temperatures.

The mechanical properties of niobium obey the general pattern for metals: with increasing temperature, the temporary resistance decreases and the relative narrowing increases. Oxygen hardens niobium and reduces its ductility. The presence of carbon impairs the mechanical properties of niobium with 0.03% oxygen [4]. In our case, we used NbStrip-1 niobium foil with a thickness of 0.2 mm. The chemical composition and mechanical properties of the foil are given in Tables 7 and 8.

**Table 7.** Chemical composition of NbStrip-1 foil in %.

| Nb | Ta | Fe | Si | O | C | N | H |
|---|---|---|---|---|---|---|---|
| main | <0.25 | <0.088 | <0.088 | <0.022 | <0.021 | <0.014 | <0.0009 |

**Table 8.** Mechanical properties of NbStrip-1 foil.

| Temperature (K) | $\sigma$ (MPa) | $\sigma_{0,2}$ (MPa) | $\delta$ (%) |
|---|---|---|---|
| 293 | 493 | 383 | 27 |

M1 grade copper is mainly used for the manufacture of electric current conductors, rolled products, and tin-free high-quality bronzes [4]. The spatial lattice of copper is face-centered cubic. The composition of the M1 copper foil is shown in Table 9.

**Table 9.** Chemical composition of M1 copper foil in %.

| Cu | Bi | Sb | As | Ni | Sn | Zn | Fe | Pb | O | Ag |
|---|---|---|---|---|---|---|---|---|---|---|
| >99.9 | <0.002 | <0.002 | <0.002 | <0.002 | <0.002 | <0.005 | <0.005 | <0.005 | <0.05 | <0.003 |

Table 10 shows the mechanical properties of the soft M1 copper used in experiments.

**Table 10.** Mechanical properties of M1 copper.

| Indicator | Designation | The Quantity |
|---|---|---|
| Melting point (K) | $T_{mel}$ | 1356 |
| Specific gravity (kg/m$^3$) | $\gamma$ | 8952 |
| Elastic modulus (GPa) | E | 132 |
| Tensile Strength (MPa) | $\sigma_{ten}$ | 227 |
| Optimal elongation (%) | $\delta$ | 60 |

In addition, with the counter diffusion of titanium and copper, there is a danger of the formation of a low-melting eutectic as a result of their interaction, if the depth of their

diffusion overlaps, which occurs when the thickness of the niobium insert is less than 20 µm. A similar picture is observed in copper foil due to the diffusion of iron into copper and niobium into copper (up to 10 µm): the formed solid solutions of iron (chromium, nickel) in copper throughout the entire volume of a copper insert with a thickness of 20–50 µm strengthen it and make it less plastic.

When the thickness of the copper foil is less than 0.4 mm, there is a danger of instability in the strength properties due to the convergence of the strengthened zones of solid solutions. An increase in the thickness of the copper foil over 0.6 mm is impractical due to a possible decrease in the strength of the titanium-steel bond. Thus, the minimum thickness of the copper layer, where diffusion processes are not observed, plasticity is preserved, and stability of strength properties is ensured, is 0.29 mm.

A double niobium gasket on the titanium side and a copper gasket on the steel side was adopted. Reducing the thickness of the copper layer to less than 0.1 mm increases the tensile strength of the joint, which is explained by the manifestation of the effect of contact hardening. Fracture of joints during testing proceeds along the copper layer and has a viscous character at positive and negative temperatures.

The technological process tests for obtaining an integral titanium-steel connection were carried out on specimen duplicates. Figure 2 shows before welding, and Figure 3 shows the welded specimens. Twenty-two No. 4 standard specimens were turned from the welded specimens. Interstate Standard 1497–84 [30] was used for the mechanical tests (10 mm in diameter and 90 mm long) as is shown in Figure 4.

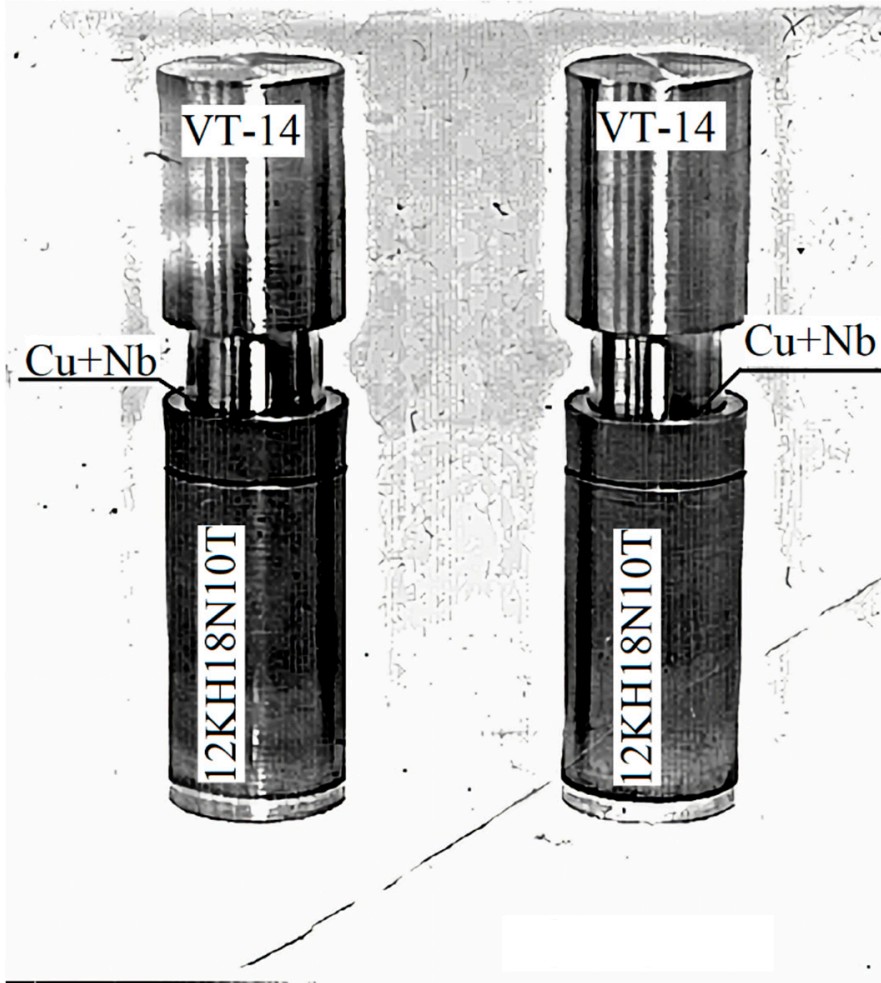

**Figure 2.** The 12KH18N10T+Cu+Nb+VT-14 sample before welding.

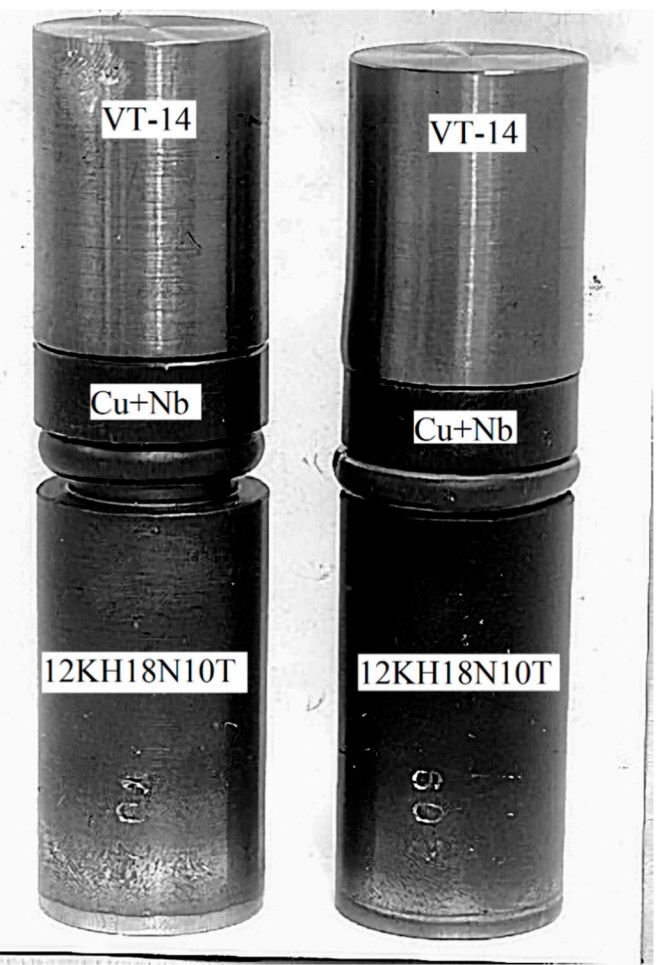

**Figure 3.** The 12KH18N10T+Cu+Nb+VT-14 welded samples.

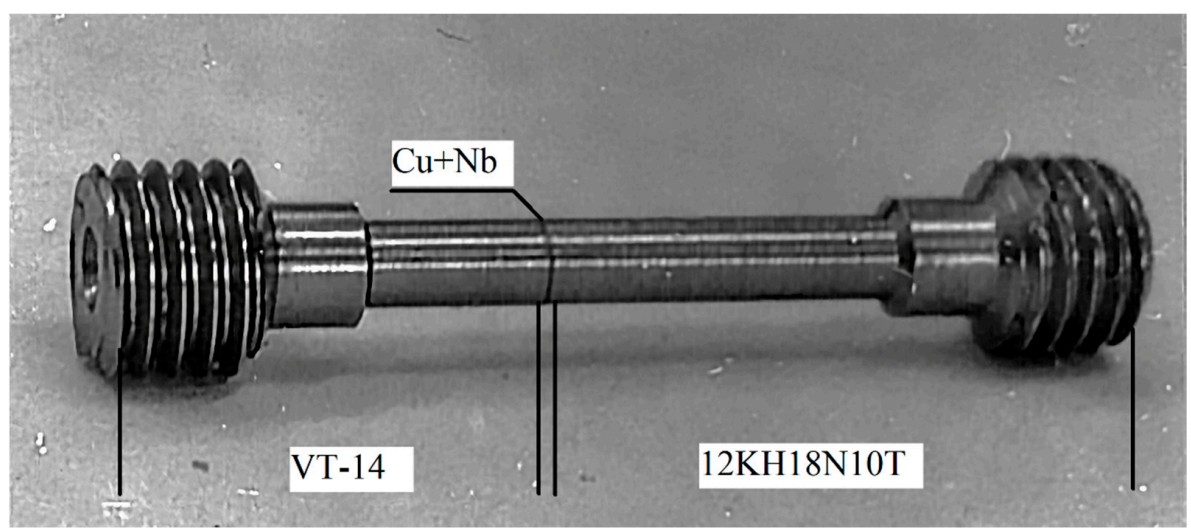

**Figure 4.** Standard sample No. 4 for mechanical tests turned from a 12KH18N10T+Cu+ Nb+VT-14 billet.

*Testing Methods*

The experiments were carried out on a A306.04 diffusion-vacuum welding unit for dissimilar materials, as shown in Figure 5.

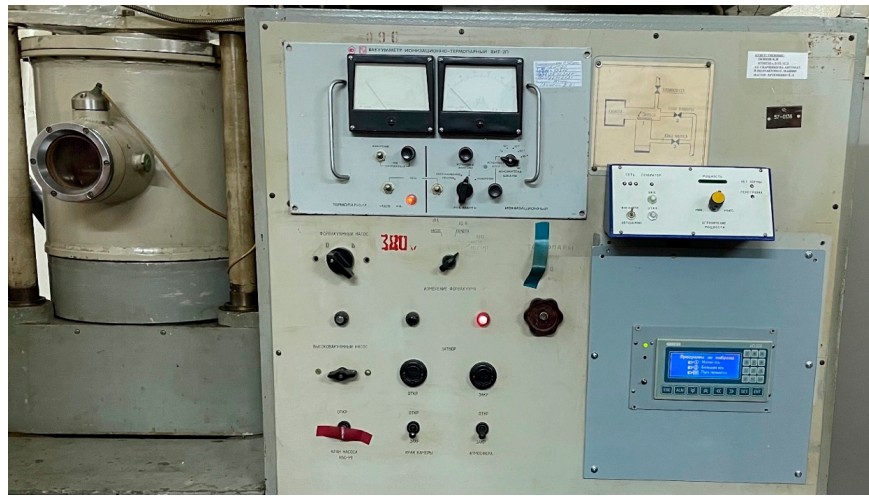

**Figure 5.** Installation for diffusion welding A306.04.

For welding the 12KH18N10T+Cu+Nb+VT-14 sample, heating was carried out on a device with an induction heater from a high-frequency LZ-2-67generator.

As shown in Figure 2, samples for welding were collected in a welding attachment, installed in a vacuum chamber, and process pressure was applied. The pressure on the specimens to be welded was transmitted through the rod and the ball joint to avoid skewing. The chromel-alumel thermocouple junction was directly introduced into the contact zone of the steel sample with spacers and titanium alloy. The diagram of the diffusion welding process is shown in Figure 6.

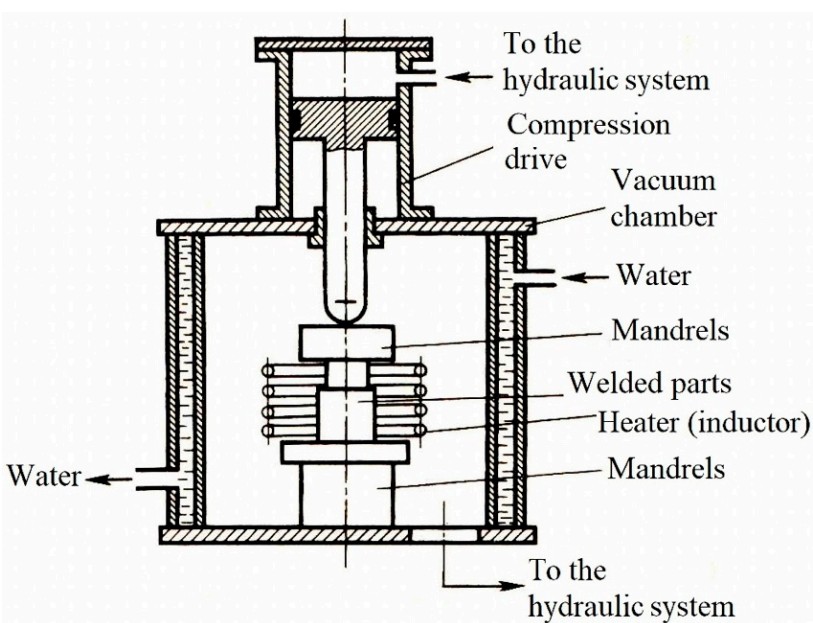

**Figure 6.** Diffusion welding scheme.

Temperature readings were taken with a KSP-3 potentiometer with an accuracy of $\pm$ 5K. The vacuum in the chamber was controlled with a VIT-2A vacuum gauge. Welding was carried out in the following sequence: a vacuum was created in the vacuum chamber (H-1.3 Pa), the samples were heated to the welding temperature ($T_w$), the welding pressure was applied (P) and held during the welding time ($\tau$), and the assembly was cooled (cooling rate ($T_{ohl}$)) to a temperature of 673 K. Then, the installation was depressurized, and the workpiece was further cooled in air.

In the course of the research, the values of the following technological parameters were determined: sample heating rate, compressive force value, material welding temperature, isothermal retention time, as well as the rate of cooling the device in air. When calculating the modes of welding titanium with steel, the strength characteristics of the materials obtained were investigated using cylindrical specimens.

The preparation of the surfaces to be joined immediately before welding is an important part of the technological process.

The welding surfaces of the VT-14 titanium and stainless steel were polished with a roughness of Ra-2.5.

Copper and niobium foil interlayers were formed by stamping. To remove oxide films, foils were processed with L154S6N Russian State Standard 10054–82 sandpaper [31], degreased with acetone, and wiped off with a lint-free cloth. The following procedure was adopted for the preparation of workpiece surfaces to be welded and the process of diffusion welding of bimetallic adapters:

1.  Perform an incoming inspection of the samples to be welded and the foil. Check the accompanying documentation.
2.  Clean the surfaces of the parts and foil to be welded.
3.  Rinse and degrease with acetone, dehydrate with rectified alcohol, and rub parts with a napkin.
4.  Quality control of the preparation of the surfaces to be welded.
5.  Assemble the parts into the welding fixture. To exclude the weldability of parts with a device, install layered gaskets between the contacting surfaces. Install the part in the installation chamber. Close the installation chamber. Apply a process pressure of 3 MPa.
6.  Evacuate the installation chamber to at least 1.3 Pa.
7.  Weld the workpieces in the following modes:

    - Heating: 1137 K;
    - Specific pressure: 18 MPa;
    - Welding time: 1200 s;
    - Vacuum: 1.3 Pa.

8.  Cool the parts under pressure in a vacuum to a temperature of 673 K.
9.  Depressurize the installation chamber and remove the welding pressure.
10. Remove the welded bimetallic part from the welding device.
11. Control the deformation of the part.
12. Ultrasonically inspect the welded part.
13. Pack the welded part and transport it to the mechanical site.

During the development of the technological process for obtaining a bimetallic adapter from the workpiece, special samples were made. To control the quality of welding, a sample for mechanical and ultrasonic tests was obtained from the welded assembly, as shown in Figure 7.

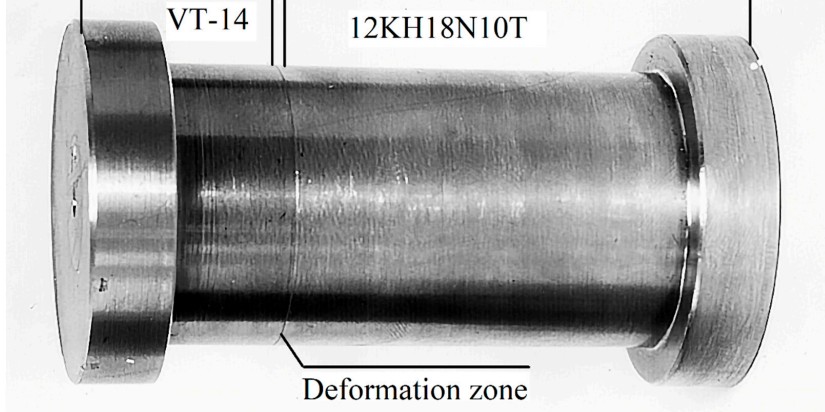

**Figure 7.** Sample for mechanical testing of the 12KH18N10T+Cu+Nb+VT-14 bimetallic adapter.

The control of the continuity of the layers was carried out from the side of titanium or steel on the samples of Figure 6, using the ultrasonic echo method with electroencephalographic (EE) sensitivity by a separate-combined direct transducer with a frequency of 5 MHz on UD brand flaw detectors (UD2–12). The contact liquid is water.

Thus, the use of ultrasonic testing of welded samples made it possible to exclude the use of rejected VT-14 + 12KH18N10T bimetallic assemblies obtained by diffusion welding (Table 11).

**Table 11.** Results of ultrasonic testing of the connection 12KH18N10T+M1+NbStrip-1+VT-14.

| No. | 31 | 32 | 33 | 34 | 35 | 36 | 37 | 38 | 52 | 58 | 59 |
|---|---|---|---|---|---|---|---|---|---|---|---|
| Results of ultrasonic | - | - | + | + | + | + | + | + | - | - | - |
| MPa | 180 | 330 | 305 | 370 | 360 | 365 | 350 | 225 | 240 | 260 | 320 |

The results of ultrasonic testing of welded joints are given in Table 11. The sign "-" indicates the presence of defects larger than $1 \times 1$ mm.

The same table shows the results of mechanical tests of these samples, which showed that the required joint strength of more than 300 MPa is provided in the absence of defects.

For carrying out cyclic tests, cylindrical specimens of a solid cross-section with a plane of connection of layers normal to the specimen axis were made. The diameter of the test section was 6 mm, the length was 17 mm, and the roughness Ra was 0.62 μm. The tests were carried out on a Schenk RSA-10 machine at a frequency of 40 Hz and a temperature of 293 K. The nature of sample loading was soft harmonic tension, and the asymmetry of the cycle was 0.05. A total of 21 samples were tested, as selected from 12 blanks. The tests were carried out under the following maximum voltage conditions: 420; 350; 300; 250; 225; 200; 150 MPa; at each stress level, 2–4 samples were tested. The bond strength of the layers under static loading was determined using samples of the same configuration at a temperature of 293 K on a Schenk RSA hydraulic machine and at an active gripping speed of 1 mm/min. Three samples were tested.

On samples that were destroyed after static and cyclic tests, the nature of destruction was analyzed by visual inspection, optical metallography, and electron fractography. All destroyed samples were subjected to visual inspection of fractures without the use of magnifying devices and under a microscope at a magnification of up to 200 times; two samples were analyzed on a JSM-35 electronic analyzer, focusing on the analysis of elements on the fracture surface.

For the impact tests, samples with a square cross-section of $5 \times 5$ mm and a length of 30 mm were made without a notch with the plane of connection of the layers normal to the sample axis. The tests were carried out at 293 K according to a two-support scheme on a pendulum tester. Ten samples were tested. After the tests, the analysis of the nature of destruction was carried out by visual inspection of fractures, optical metallography, and electronic fractography. All of the samples were subjected to visual inspection, three samples were subjected to optical metallography, and one sample was subjected to electronic fractography.

To determine the oxygen content, 10 samples, each $8 \times 5 \times 5$ mm in size, were fashioned by milling from the initial titanium billet and the titanium component. The analysis was carried out using the vacuum-melting method with a metal bath (iron) at a temperature of 2073 K. The arithmetic mean of the two parallel determination results was taken as the final analysis result.

To determine the thicknesses of the intermediate layers and analyze the nature of the fractures, metallographic thin sections were etched on the surface, normal to the plane of the layer join. Sections were prepared by mechanical grinding and polishing. To identify the boundaries, a mixture of nitric and hydrofluoric acids in the ratio (1:1) was used for etching the titanium component. The analysis of the thin sections was carried out on an MMU-3 metallographic microscope with a magnification of up to 200 times. The thickness of the intermediate layers was changed at a magnification of 300 times using the method of

random sections, using the eyepiece of an MOV-1-15 micrometer calibrated according to the OMO U4.2 standard ruler with a graduation value of 0.01 mm. The thickness of the layers was determined as the arithmetic mean of 10 measurements.

## 3. Results and Discussion

During the research, gaskets made of niobium on the titanium side and copper on the steel side were used. Heating the joints up to 1073 K using barrier subcoats did not lead to weld embrittlement. The decrease in the ultimate strength was associated with the removal of the work-hardening effect. Reducing the thickness of the copper layer to 0.1 mm increased the tensile strength of the joint, which is explained by the contact hardening effect. The destruction of the connections during the tests passed through the copper layer and had a viscous character at positive and negative temperatures (+573–173K).

The results of the mechanical tests of samples under static and cyclic loading are shown in Table 12.

**Table 12.** Cyclic test results.

| No. Sample | Max Level Stresses (MPa) | Number Cycles (Times) | Failure Location |
|---|---|---|---|
| 1 | 510 | Static strength | At both boundaries of the copper interlayer: Nb-Cu and Cu-12KH18N10T |
| 2 | 522 | Static strength | At both boundaries of the copper interlayer: Nb-Cu and Cu-12KH18N10T |
| 3 | 517 | Static strength | At both boundaries of the copper interlayer: Nb-Cu and Cu-12KH18N10T |
| | | Min-max 510–526 Avg. 516 | |
| 1 | 400 | 0.7e4 | At both boundaries of the copper interlayer: Nb-Cu and Cu-12KH18N10T |
| 2 | 400 | 0.8e4 | At both boundaries of the copper interlayer: Nb-Cu and Cu-12KH18N10T |
| 3 | 400 | 1e4 | At both boundaries of the copper interlayer: Nb-Cu and Cu-12KH18N10T |
| 1 | 350 | 1.6e4 | At both boundaries of the copper interlayer: Nb-Cu and Cu-12KH18N10T |
| 2 | 350 | 2.6e4 | At both boundaries of the copper interlayer: Nb-Cu and Cu-12KH18N10T |
| 3 | 350 | 2.8e4 | At both boundaries of the copper interlayer: Nb-Cu and Cu-12KH18N10T |
| 1 | 300 | 3.4e4 | At both boundaries of the copper interlayer: Nb-Cu and Cu-12KH18N10T |
| 2 | 30.0 | 7.4e4 | At both boundaries of the copper interlayer: Nb-Cu and Cu-12KH18N10T |
| 3 | 30.0 | 10.8e4 | At both boundaries of the copper interlayer: Nb-Cu and Cu-12KH18N10T |
| 4 | 30.0 | 12.1e4 | At both boundaries of the copper interlayer: Nb-Cu and Cu-12KH18N10T |
| 1 | 250 | 22.5e4 | At both boundaries of the copper interlayer: Nb-Cu and Cu-12KH18N10T |
| 2 | 250 | 42.9e4 | At both boundaries of the copper interlayer: Nb-Cu and Cu-12KH18N10T |
| 3 | 250 | 43.6e4 | At both boundaries of the copper interlayer: Nb-Cu and Cu-12KH18N10T |
| 4 | 250 | 45.6e4 | At both boundaries of the copper interlayer: Nb-Cu and Cu-12KH18N10T |
| 1 | 225 | 85.9e4 | At both boundaries of the copper interlayer: Nb-Cu and Cu-12KH18N10T |
| 2 | 225 | 137.7e4 | At both boundaries of the copper interlayer: Nb-Cu and Cu-12KH18N10T |
| 1 | 200 | 200e4 | Samples did not collapse |
| 2 | 200 | 200e4 | Samples did not collapse |
| 1 | 150 | 200e4 | Samples did not collapse |
| 2 | 150 | 200e4 | Samples did not collapse |
| 3 | 150 | 200e4 | Samples did not collapse |

The values of the logarithms of the sample durability obtained during the tests, their arithmetic mean values, and standard deviations depending on the stress level are shown in Figure 8.

We should note that specimens with durability logarithms of 6.301 were not destroyed during the tests.

The endurance limit based on 2000 cycles, with a probability of destruction tending to zero, was 147 MPa, and with a probability of 50%, was - 171 MPa. The control points of the fatigue curve, as calculated for a fracture probability of 50% and dependent on the level of maximum stress, are given in Figure 9.

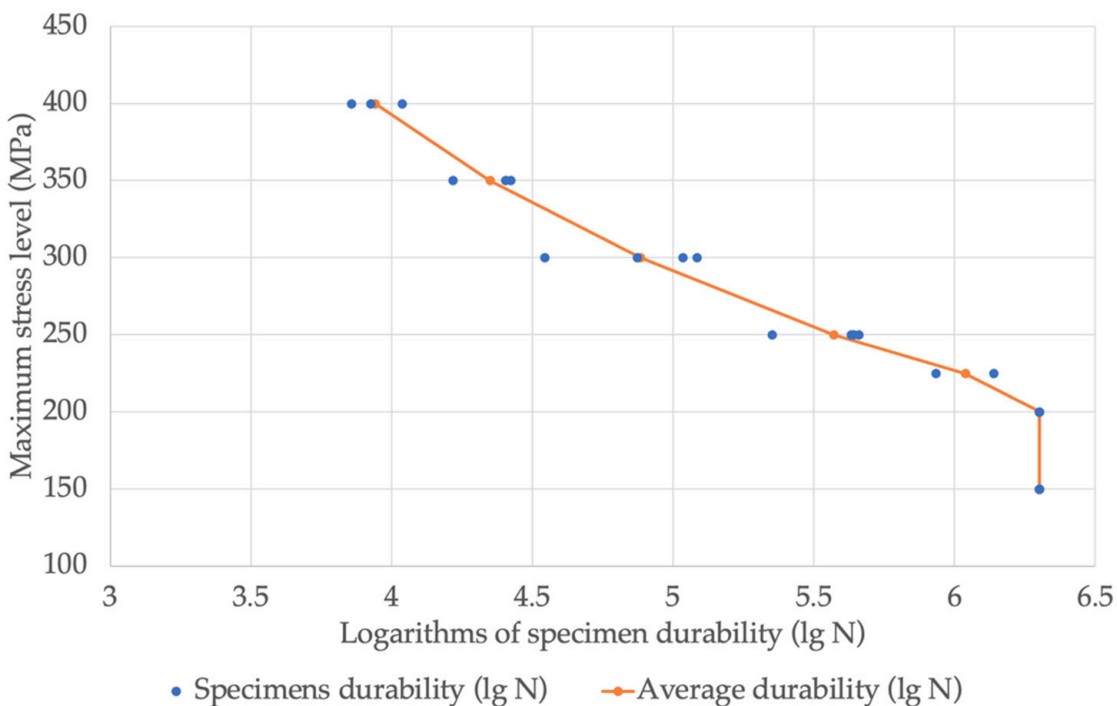

**Figure 8.** Durability of VT-14+Nb+Cu+12KH18N10T specimens obtained by diffusion welding during cyclic tests.

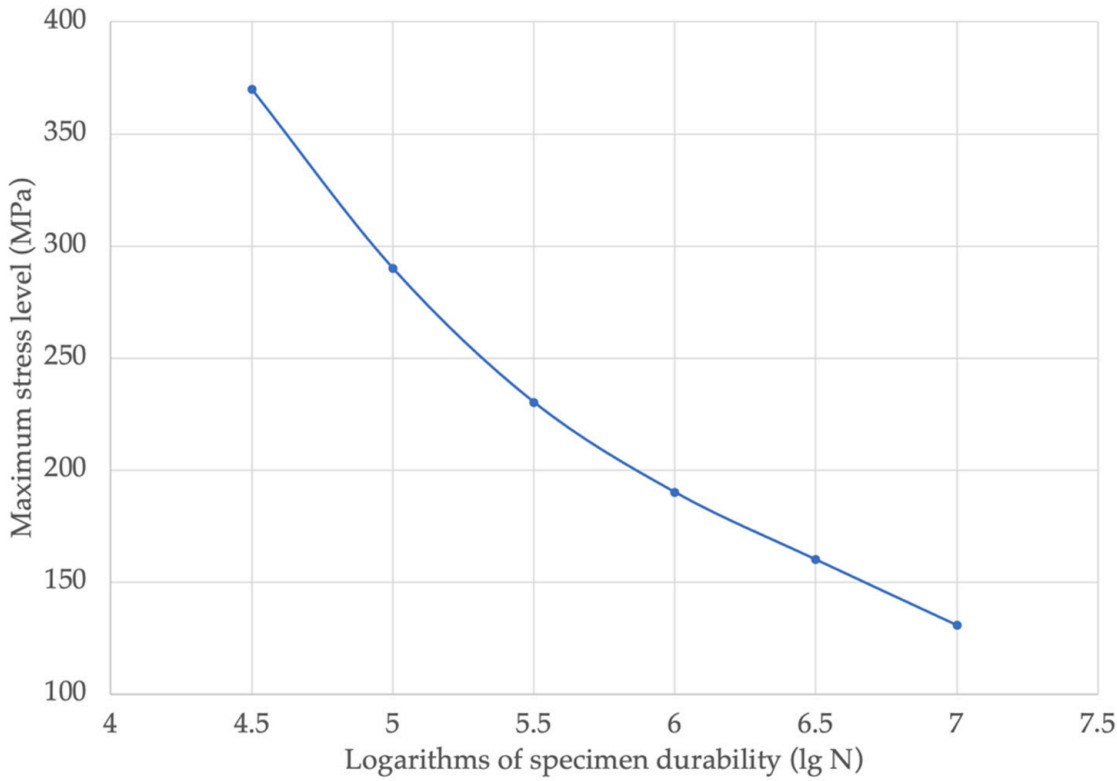

**Figure 9.** Calculated values of the fatigue curve of VT-14+Nb+Cu+12KH18N10T specimens obtained by diffusion welding with a fracture probability of 50%.

The impact strength results and the analysis of the nature of the fractures are shown in Table 13.

**Table 13.** Impact strength test results.

| Sample No. | Destruction Work (kgf.m/cm) | Failure Location |
|:---:|:---:|:---:|
| 1 | 1.1 | At both boundaries of the copper interlayer: Nb-Cu and Cu-12KH18N10T |
| 2 | 1.5 | At both boundaries of the copper interlayer: Nb-Cu and Cu-12KH18N10T |
| 3 | 0.88 | At both boundaries of the copper interlayer: Nb-Cu and Cu-12KH18N10T |
| 4 | 1.1 | At both boundaries of the copper interlayer: Nb-Cu and Cu-12KH18N10T |
| 5 | 1.6 | At both boundaries of the copper interlayer: Nb-Cu and Cu-12KH18N10T |
| 6 | 1.0 | At both boundaries of the copper interlayer: Nb-Cu and Cu-12KH18N10T |
| 7 | 0.64 | At both boundaries of the copper interlayer: Nb-Cu and Cu-12KH18N10T |
| 8 | 1.2 | At both boundaries of the copper interlayer: Nb-Cu and Cu-12KH18N10T |
| 9 | 1.1 | At both boundaries of the copper interlayer: Nb-Cu and Cu-12KH18N10T |
| 10 | 1.1 | At both boundaries of the copper interlayer: Nb-Cu and Cu-12KH18N10T |
| **Min** | 0.64 | - |
| **Max** | 1.6 | - |
| **Avg** | 1.12 | - |

Visual inspection after the mechanical tests to assess the nature of destruction found that both fracture surfaces were red (copper-plated), there was a lack of fusion due to nonresident inclusions, and there was no oxidation of the contact surfaces. However, in comparison with the fractures of the samples obtained by vacuum rolling, the fractures exhibited a gray tint. The analysis of the fracture site on the metallographic thin sections showed that fractures from all types of tests (static, cyclic, shock) occurred along the boundaries of niobium-copper and copper-steel joints. Moreover, a copper interlayer was found at the fracture site either on the niobium or steel surface, as shown in Figure 10.

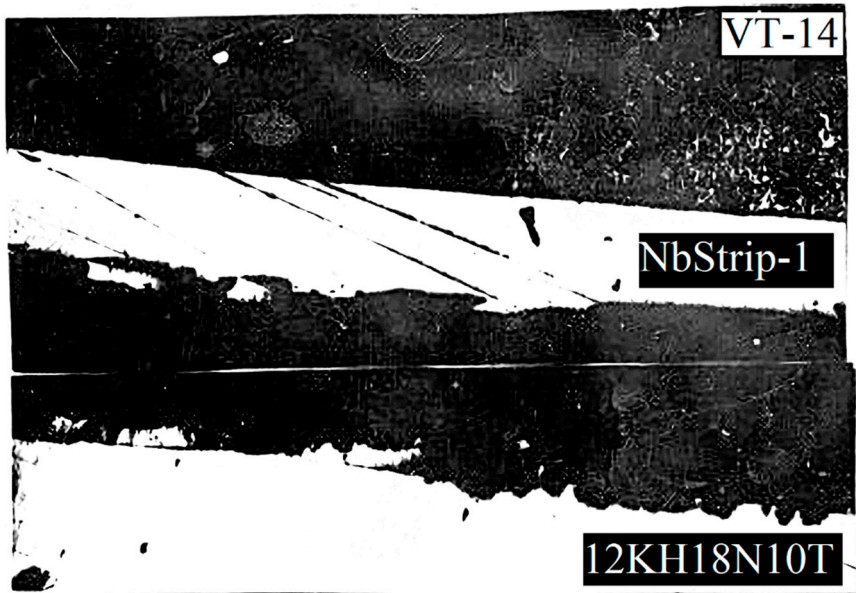

**Figure 10.** View of the destroyed sample after testing.

Using electronic fractography, a dimple structure was observed on the fracture surfaces, and various elements (Fe, Cr, Nb) were found on the protrusions and depressions, which are part of the sublayer (steel or niobium) that followed the copper, as shown in Figure 11.

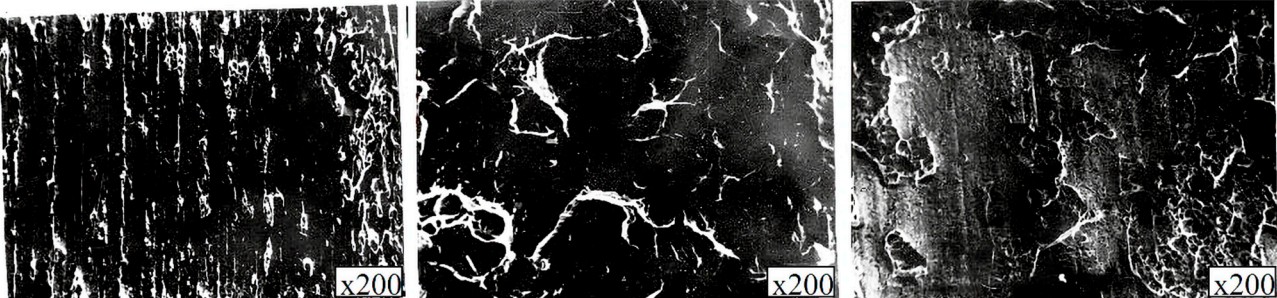

**Figure 11.** Fractography of the sample surface after destruction.

The chemical analysis results of the oxygen content in the titanium component of the bimetal before and after diffusion are shown in Table 14.

**Table 14.** Oxygen content in titanium alloy before and after diffusion welding.

| Metal Condition | Molecular Oxygen Content (%) |
|---|---|
| Initial | 0.07 |
| After diffusion welding | 0.07 |

The thickness measurements of the intermediate layers of niobium and copper, performed on thin sections of the bimetallic VT-14+Nb+Cu+12KH18N10T compound, showed that the average thickness of niobium was 0.2 mm, and the average thickness of copper was 0.02 mm. The niobium and copper interlayers were solid, and no breaks were observed, as shown in Figure 12.

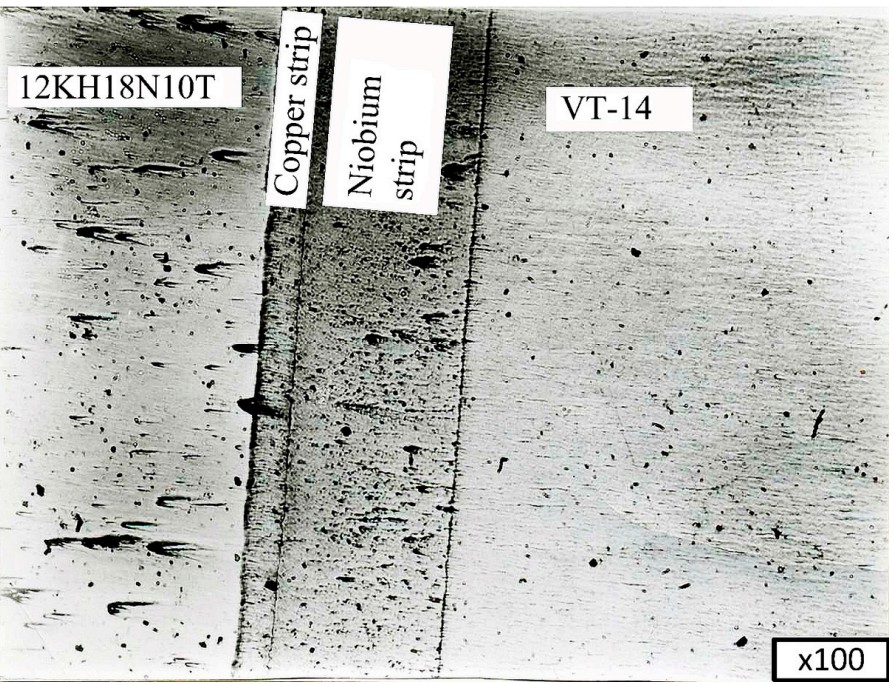

**Figure 12.** Microsection of welded solid-phase VT-14+Nb+Cu+12KH18N10T joint.

Samples for tensile tests were pumped in from the welded samples, and the final groove of the samples to a diameter of 10 mm and 35 mm must be performed from titanium to steel.

The first cycle of tensile tests was carried out on an RSM–50 tensile testing machine at a speed of 10 mm/min at a temperature of 293 K. The samples were tested at high and low temperatures.

The maximum bond strength at 293 K in various experiments reached 320–350 MPa, which meets the technical requirements. Furthermore, it was found that in all cases, the destruction of the samples occurred over the entire area of the sample, along the copper interlayer (NbStrip-1), as shown in Figure 10.

A maximum joint strength of $\sigma_{ten}$ = 350 MPa was obtained with a niobium foil thickness of 0.2 mm in the following welding modes: $T_b$ = 1137 K, $P_c$ = 18 MPa, and T = 1200 s. By increasing the thickness of the NbStrip-1 foil strip to 0.3 mm, the strength of the joint obtained in the same welding modes increased to 475—535 MPa.

The samples were also subjected to tensile tests after heating three times. Thus, the results of these tests showed that the strength of the bimetallic workpieces, obtained by diffusion welding at temperatures of 77, 273, 373, and 473 K, and after heating to 673 K three times, was more than 320 MPa, and the destruction occurred along the copper interlayer over the entire area of the sample.

To study the microstructure of the joints and determine the thickness of the niobium and copper gaskets, metallographic studies were carried out. Microsections were prepared from VT-14+Nb+Cu+12KH18N10T welded joints to carry out these investigations. The studies were conducted on an MIM-7optical microscope. Figures 13–15 show microsections of the VT-14+Nb+Cu+12KH18N10T joint obtained by diffusion welding at various magnifications.

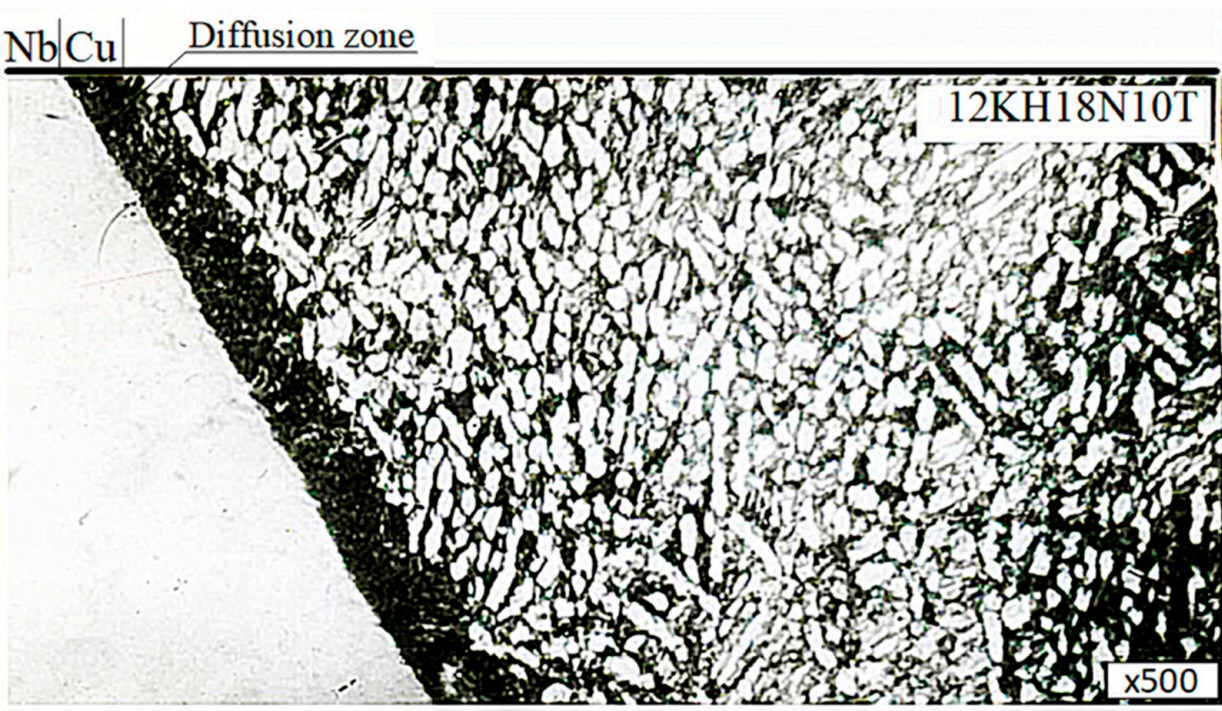

**Figure 13.** Connection area of 12KH18N10T stainless steel and copper.

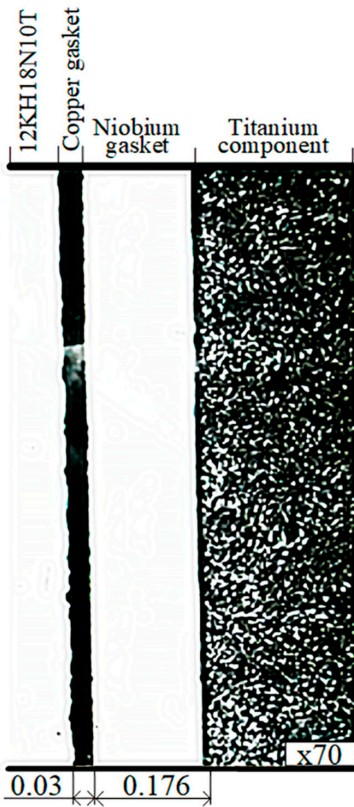

**Figure 14.** VT-14+Nb+Cu+12KH18N10T connection macrosection.

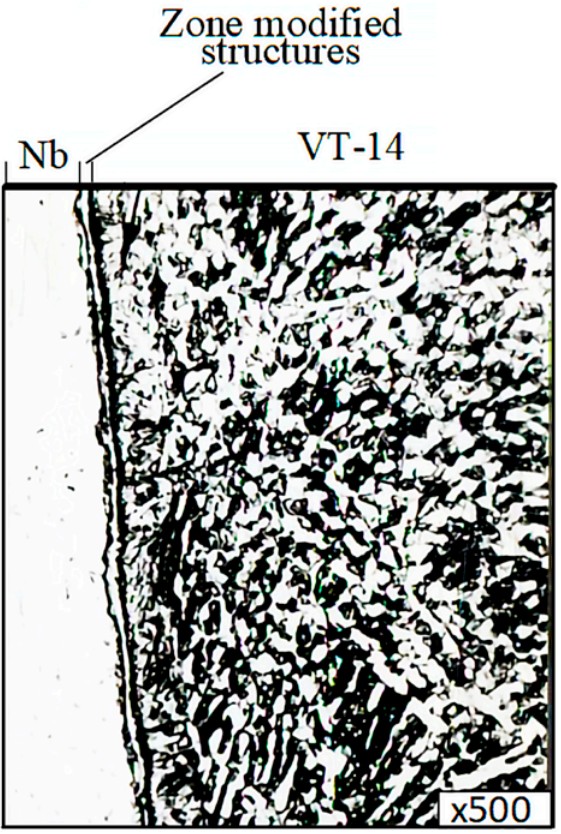

**Figure 15.** VT-14+Nb connection macrosection.

The results showed that the thickness of the niobium layer was 0.18–0.19 mm, and the thickness of the copper layer was 0.04–0.045 mm.

The diffusion zone at the 12KH18N10T + M1 and niobium + copper boundary was not revealed by metallographic studies; for this, it was necessary to select the appropriate thin section preparation. However, the presence of this a zone was confirmed in works [1]. Moreover, the predominant character of intercrystalline diffusion was noted. The most complete study of steel + copper bimetal can be found in [2,3].

In addition, it was found that plastic deformation contributes to the flow of diffusion processes. The strength of the diffusion layer was higher than that of copper and steel.

The presence of diffusion zones at the steel + copper and copper + niobium interface was confirmed by the tensile tests results, which showed that the destruction of the samples in all cases occurred over the entire area of the samples over the less durable material, i.e., copper. To ensure a rupture in the copper, the thickness of the interlayer must be more than 15 microns, while the diffusion depth compensates for the insufficient contact of the contacting surfaces.

Microhardness measurement was used to study the properties of the contact zone. The microhardness was measured on a PMT-3 device at a load of 0.05 kg.

The results showed that the microhardness of the titanium core was equal to 301–312 HV. The microhardness of the niobium pad was 112–101 HV. The microhardness of the transition zone had an intermediate value of 270–186 HV (Figure 16), which was also confirmed the diffusion interaction of the contacting metals.

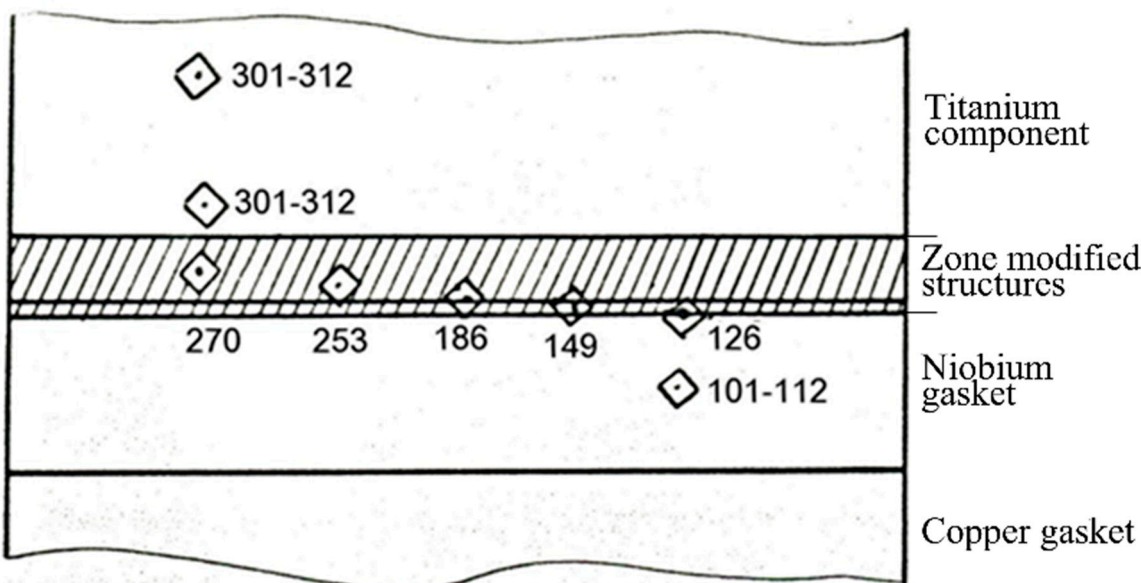

**Figure 16.** Distribution of microhardness in the 12KH18N10T+M1+NbStrip-1+VT-14 joint zone.

In diffusion welding, the quality of the joint is ensured in the presence of physical contact, the formation of which occurs as a result of the convergence of the surfaces to be joined due to plastic deformation of microprotrusions and surface layers.

In the three selected VT-14+NbStrip-1+M1+12KH18N10T samples, after mechanical tests, the content of gas impurities ($N_2$ and $H_2$) was determined. The results showed that the nitrogen content in all samples was 0.01 and the hydrogen content was 0.005-0.006%. A hydrogen content of 0.006% was observed in one sample, which was analyzed after heating three times to a temperature of 673 K.

A special prepared sample, cut from a part, was tested with a helium-air mixture under a pressure of 3600 MPa with a helium concentration in the test gas of at least 40%. The results regarding the leakage of the helium mixture showed a high-quality VT-14+NbStrip-1+M1+12KH18N10T compound.

Gasket measurements from the metallographic analysis of the adapter for the welded bimetallic samples showed the following: niobium: 0.18–0.19 mm; copper: 0.04–0.045 mm.

Titanium diffuses into niobium to a depth of 40 microns (Figure 17), and copper diffuses into niobium up to 10 microns (Figure 18), forming, in both cases, solid solutions throughout the entire volume of a niobium insert 20–50 microns thick, which strengthen it and make it less plastic.

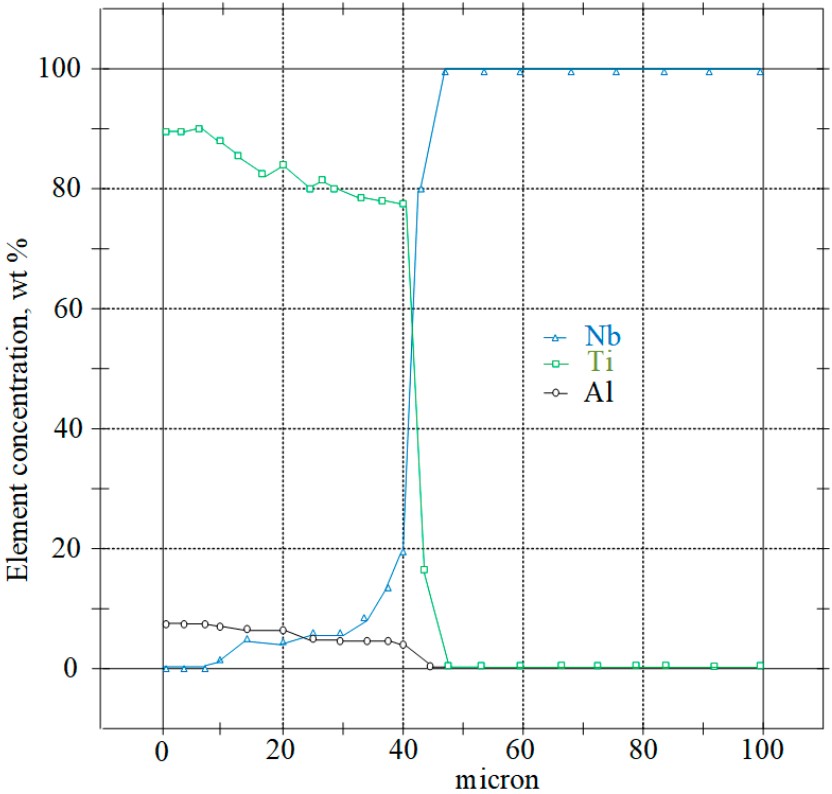

**Figure 17.** Diffusion welding scheme 12KH18N10T+Cu+Nb+VT-14.

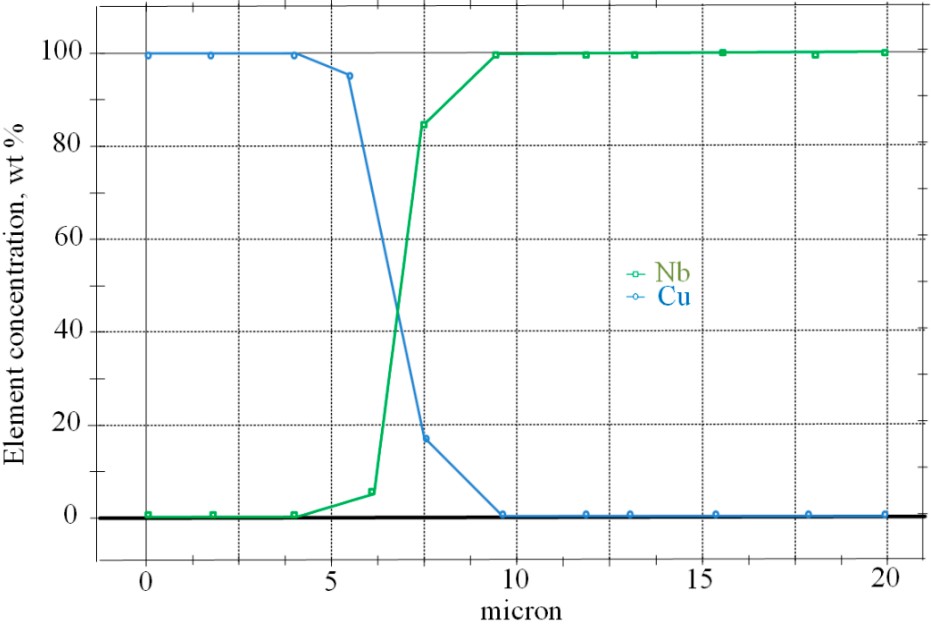

**Figure 18.** Diffusion welding scheme 12KH18N10T+Cu+Nb+VT-14.

Figure 18 shows a diagram of the distribution of the main alloying elements across the zone of diffusion copper-steel connection.

At the interface between niobium and titanium, zones of solid solutions with increased hardness are formed. When choosing the thickness of the niobium interlayer 0.2–0.4 mm, the thickness of the copper gasket will be selected accordingly 0.1–0.15 mm. Increasing the thickness of the niobium foil over 0.4 mm is impractical due to a possible decrease in the strength of the titanium-steel adapter due to the absence of the "soft interlayer effect". Thus, the minimum thickness of the niobium layer, where diffusion processes are not observed and plasticity is retained, is 0.2 mm

When the thickness of the copper foil is less than 0.4 mm, there is a danger of instability of the strength properties due to the convergence of the strengthened zones of solid solutions. An increase in the thickness of the copper foil over 0.6 mm is impractical due to a possible decrease in the strength of the titanium-steel bond. Thus, the minimum thickness of the copper layer, where diffusion processes are not observed, plasticity is preserved, and stability of strength properties is ensured, is 0.29 mm.

When heated, refractory metals (tungsten, molybdenum, niobium) interact vigorously with the gases of the surrounding atmosphere to form oxides, nitrides, and carbides. These compounds are precipitated along the grain boundaries and sharply reduce the plastic characteristics of the metal. To a lesser extent than tungsten and molybdenum, niobium and tantalum are susceptible to the embrittlement of impurities. Therefore, welding must be carried out in a vacuum.

In addition, it was found that the microstructure of the microsection titanium component consisted of a mixture of (a + b)-phases and corresponded to type 4–6 of the 9 typical scales for (a + b)-titanium alloy microstructures, according to OST 97 9465-81. At the point of contact of the thin section component in the niobium interlayer, a zone characterized by a modified structure (a diffusion zone) with a width of up to 0.011 mm was found, as shown in Figure 19.

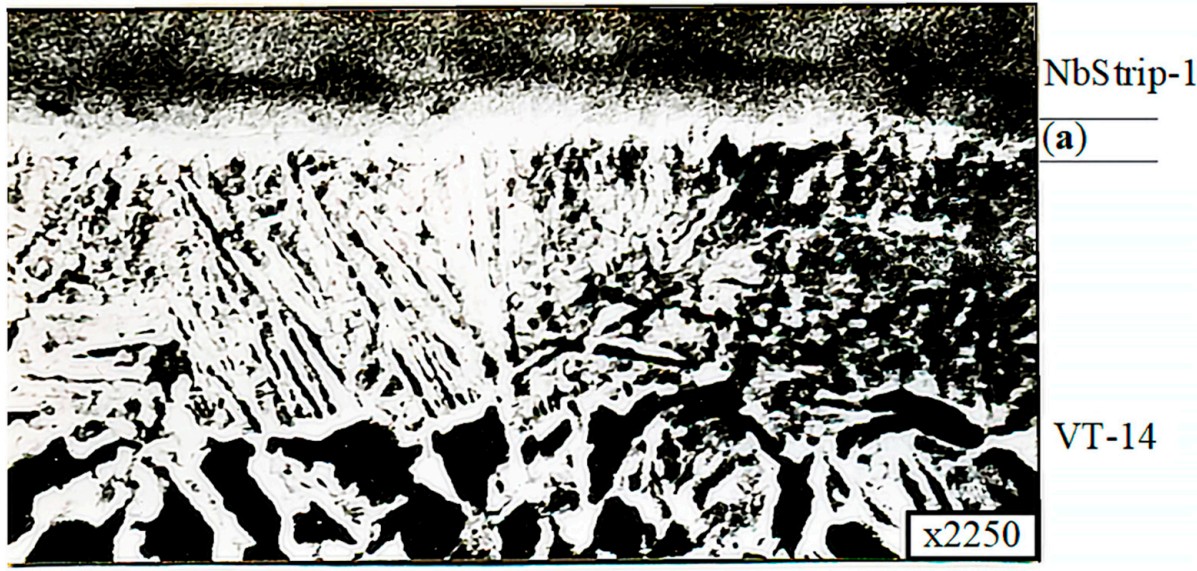

**Figure 19.** Niobium-titanium compound (secondary electrons): (**a**) the zone characterized by an altered structure.

Electron microscopic studies were carried out on microsections of the VT-14+NbStrip-1+M1+12KH18N10T welded joint. These were obtained by diffusion welding under optimal conditions. The studies were carried out on a REM-100U electron microscope. To reveal the microstructure, the metals were chemically etched with reagents. The results showed that in the NbStrip-1+VT-14 connection zone, a zone characterized by an altered structure (a diffusion zone) with a width of 0.011 mm was found (Figure 15). Moreover, a

diffusion zone was revealed at the boundary of the 12KH18N10T+M1 compound with a width of up to 0.005 mm (Figure 13).

The NbStrip-1+VT-14 transition zone was studied by X-ray microanalysis using an REMMA-202.

The results showed the presence of both titanium and niobium in the transition zone, which confirms the occurrence of diffusion processes at the boundary, as shown in Figures 19 and 20.

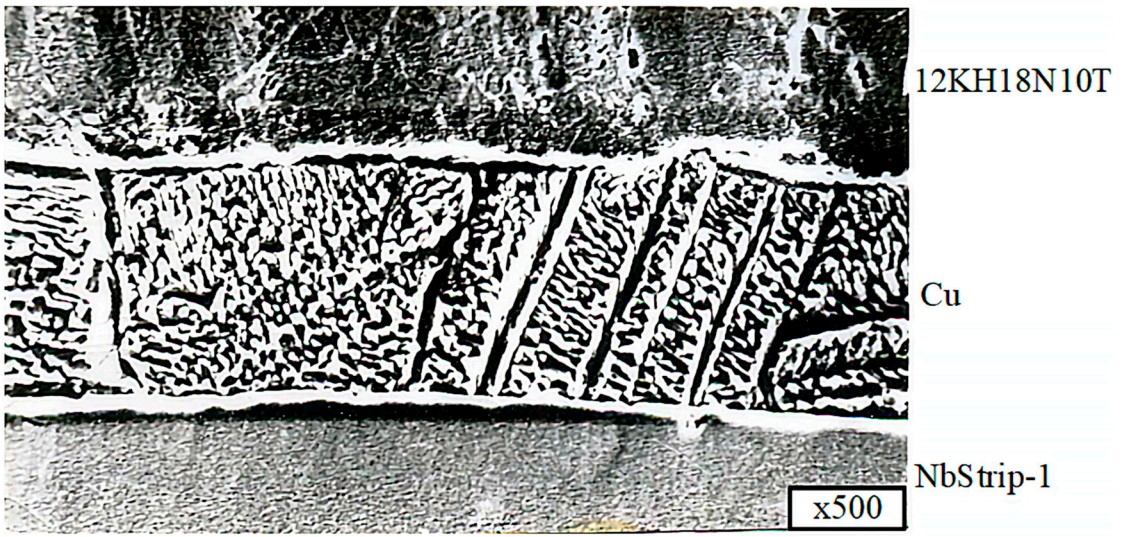

**Figure 20.** 12KH18N10T+NbStrip-1 (in secondary electrons) connection.

When the thickness of the copper foil was less than 0.4 mm, there was a danger of the strength properties becoming unstable due to the convergence of the hardened zones of solid solutions. An increase in the thickness of the copper foil to over 0.6 mm is impractical due to a possible decrease in the strength of the titanium-steel bond. Thus, at a minimum copper layer thickness of 0.3 mm, plasticity is preserved, and stability of strength properties is ensured, as diffusion processes are not observed here.

In addition, the ratio of the thickness of the copper to niobium foil should be 1.5–3 in order to initiate the destruction of the joint along the copper interlayer, which determines the predictable and stable strength properties of the joint, and its good ductility.

Thus, the results of the mechanical tests and studies show that 12KH18N10T+NbStrip-1+M1+VT-14 bimetallic billets obtained by diffusion welding in optimal modes meet the technical requirements.

## 4. Conclusions

In the course of the study, a technological process was developed for joining VT-14 titanium alloy and 12KH18N10T stainless steel using a bimetallic spacer (Cu + Nb) in the process of diffusion welding. As a result, a new technological scheme for the manufacture of bimetallic blanks using diffusion welding was proposed and experimentally investigated. The results showed that the microhardness of the niobium pad was 112–101 HV and the microhardness of the transition zone had an intermediate value of 270–186 HV (Figure 16), which was also confirmed the diffusion interaction of the contacting metals. Also, it was revealed that samples of VT-14+12KH18N10T bimetal formed by diffusion welding in a vacuum using niobium—copper meet the technical requirements for a bimetallic joint in terms of the strength of the connection of layers under static and cyclic loads, oxygen content in the titanium bimetal component, and impact strength. In addition, it was established that the diffusion connection of bimetallic fittings (12KH18N10T+Cu+Nb+VT-14 steel), obtained in the following optimal modes: $T_w$ = 1137 K; P = 18 MPa; $\tau$ = 1200 s; $V_{cohl}$ -0.1 K/s, have a bond strength of 320–350 MPa.

Thus, the fatigue resistance of the VT-14+12KH18N10T bimetal, obtained by diffusion welding using Nb and Cu, increases the ultimate strength values for the bimetal by about 25%, which is explained by the higher static tensile strength. In addition, also, during the study it was found that destruction of VT-14+12KH18N10T bimetallic joints, obtained by diffusion welding using Nb and Cu, through static tension, cyclic tensile tests, and impact toughness tests occurs at copper–niobium and copper–steel joints with rupture along the copper gaskets. In conclusion, it was revealed that the oxygen content in the titanium component during diffusion welding of the bimetal does not change according to the selected modes. Moreover, the bimetal samples, presented in terms of oxygen content, meet the technical requirements.

Future research should focus on the formation of joints through intermediate layers between other widely used alloys in the aerospace industry (alloys of brass, bronze, etc.).

**Author Contributions:** Conceptualization, A.V.L. (Alexander Viktorovich Lavrishchev), V.V.K. and V.V.T.; Data curation, S.V.P., A.V.M., K.A.B. and A.V.L. (Aleksey Vasilyevich Lysyannikov); Formal analysis, S.V.P., V.S.T., K.A.B. and V.V.T.; Investigation, A.V.L. (Alexander Viktorovich Lavrishchev), V.S.T., R.B.S. and V.V.T.; Methodology, A.V.L. (Alexander Viktorovich Lavrishchev), A.V.M., V.V.K. and K.A.B.; Project administration, V.S.T. and A.V.L. (Aleksey Vasilyevich Lysyannikov); Resources, A.V.M., V.V.K., K.A.B. and A.V.L. (Aleksey Vasilyevich Lysyannikov); Supervision, V.S.T.; Validation, S.V.P., R.B.S. and V.V.T.; Visualization, A.V.L. (Alexander Viktorovich Lavrishchev), S.V.P., A.V.M., V.V.K., R.B.S. and A.V.L. (Aleksey Vasilyevich Lysyannikov); Writing—original draft, A.V.L. (Alexander Viktorovich Lavrishchev), V.S.T., V.V.K., R.B.S., V.V.T and A.V.L. (Aleksey Vasilyevich Lysyannikov); Writing—review & editing, V.S.T., A.V.M. and K.A.B. All authors have read and agreed to the published version of the manuscript.

**Funding:** Not applicable.

**Institutional Review Board Statement:** Not applicable.

**Informed Consent Statement:** Not applicable.

**Data Availability Statement:** Not applicable.

**Conflicts of Interest:** The authors declare no conflict of interest.

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
