# Peer review of "Investigation of the Solid-Phase Joint of VT-14 Titanium Alloy with 12KH18N10T Stainless Steel Obtained by Diffusion Welding through Intermediate Layers"

_metals, doi:10.3390/met11081325_

Round 1

Reviewer 1 Report

  1. The authors have declared a number of problems to be faced when welding Ti-alloys to SS as well as the different methods to overcome such a problematic, as indicated by the provided references; however, the authors should clearly address the aim of this research work at the introduction section, particularly by pointing out the main problem to be solved.
  2. Why is so important to mention the following at the materials and methods section? “The closest material T-A4D3V in France and 4Al-3Mo-1V in USA; 159 • -1 TU 48-4-317-74 [26]; “…. Authors should explain…(if not necessary here, please move it to the introduction section)
  3. Table 1, Table 2, and the described quality requirements in page 4 are not necessary within this section, and should be either referenced or included as append…
  4. “Table 3. Chemical composition of VT-14 bar in %.” – % refers to wt % ? (clearly indicate)
  5. Figure 1 and Figure 2, clearly point out each material at the figures (use arrows and text)…
  6. Figure 4 should be provided preferably in color and mandatory in high resolution… Similarly for all figures. Images provided at Figure 8 are of unacceptable quality…
  7. Schematic in figure 5 is fuzzy, it should be improved…
  8. Table 11 and table 14, “place of destruction”; please use “failure location” instead. Besides, it is not clear if failure occurred at “Boundary Cu-Nb” or at the “Cu-12KH18N10T” side, or in both?... further in the manuscript the following statement was provided: “in all cases, the 406 destruction of the samples occurred over the entire area of the sample, along the copper 407 interlayer”, thus making it confusing…
  9. A hardness distribution graph would be very helpful by visualizing the variations along the different zones within the weldment; that should be incorporated in figure 13…
  10. The authors have stated that: “The NbStrip-1+VT-14 transition zone was studied by X-ray microanalysis using an REMMA-202” and that: “The results showed the presence of both titanium and niobium in the transition zone, which confirms the occurrence of diffusion processes at the boundary, as shown in Figures 14 and 15.”; however, the SEM images show only secondary electron technique (Figures 14 and 15), and backscattering electrons technique were not provided and or EDS line or mapping analysis was not carried out, hence, such evidence is incomplete and is not possible to clearly confirm the diffusion phenomena as mentioned previously…EDS or WDS analysis is mandatory
  11. Number the conclusions

Author Response

Dear Reviewer 1,

Thank you for your positive comments on our paper. You provided remarks about it, so we’ve made changes in the paper.

Remark 1: The authors have declared a number of problems to be faced when welding Ti-alloys to SS as well as the different methods to overcome such a problematic, as indicated by the provided references; however, the authors should clearly address the aim of this research work at the introduction section, particularly by pointing out the main problem to be solved.

Reply: Thank you for the comment. We added the aim of this research and clearly describe the problem. Lines 42-61, 84-90.

Remark 2: Why is so important to mention the following at the materials and methods section? “The closest material T-A4D3V in France and 4Al-3Mo-1V in USA; 159 • -1 TU 48-4-317-74 [26]; “…. Authors should explain…(if not necessary here, please move it to the introduction section).

Reply: Thank you for the comment. We moved it to the introduction (lines 177-184).

Remark 3: Table 1, Table 2, and the described quality requirements in page 4 are not necessary within this section, and should be either referenced or included as append…

Reply: Thank you for the comment.

Remark 4: “Table 3. Chemical composition of VT-14 bar in %.” – % refers to wt % ? (clearly indicate).

Reply: Thank you for the comment. We changed % to wt %. Line 217.

Remark 5: Figure 1 and Figure 2, clearly point out each material at the figures (use arrows and text).

Reply: Thank you for the comment. We added text (which material is used) at the figures.

Remark 6: Figure 4 should be provided preferably in color and mandatory in high resolution… Similarly for all figures. Images provided at Figure 8 are of unacceptable quality.

Reply: Thank you for the comment. Figure 4 is color and it is a real equipment (line 287). We changed Figure 8 to better quality (line 427).

Remark 7:  Schematic in figure 5 is fuzzy, it should be improved.

Reply: Thank you for the comment. We improved it.

Remark 8: Table 11 and table 14, “place of destruction”; please use “failure location” instead. Besides, it is not clear if failure occurred at “Boundary Cu-Nb” or at the “Cu-12KH18N10T” side, or in both?... further in the manuscript the following statement was provided: “in all cases, the 406 destruction of the samples occurred over the entire area of the sample, along the copper 407 interlayer”, thus making it confusing.

Reply: Thank you for the comment. We changed phrase “place of destruction” to “failure location” at Table 11 and Table 12 (lines 412, 431). It is not a copper, it is the alloy of Cu and Nb, we added explanation to sentences “in all cases, the…” (lines 469-471). In all experiments in which rupture occurred, the elements of the article were separated by the Cu interlayer. At the same time, the remnants of such an interlayer were found both at the border with niobium and at the border of stainless steel, so in table 12 and 13 we made a clarification in the last column “At both boundaries of the copper interlayer: Nb-Cu and Cu-12KH18N10T”

Remark 9: A hardness distribution graph would be very helpful by visualizing the variations along the different zones within the weldment; that should be incorporated in figure 13.

Reply: Thank you for the comment. Measurement of the microhardness of the titanium and niobium joint and the transition zone between them is shown in Figure 16. Figure 14 shows the optical dimensions of the copper and niobium spacers after diffusion welding. Considering that the thickness of the copper spacer after diffusion welding was 0.01-0.03, microhardness measurements were not carried out at these thicknesses.

Remark 10: The authors have stated that: “The NbStrip-1+VT-14 transition zone was studied by X-ray microanalysis using an REMMA-202” and that: “The results showed the presence of both titanium and niobium in the transition zone, which confirms the occurrence of diffusion processes at the boundary, as shown in Figures 14 and 15.”; however, the SEM images show only secondary electron technique (Figures 14 and 15), and backscattering electrons technique were not provided and or EDS line or mapping analysis was not carried out, hence, such evidence is incomplete and is not possible to clearly confirm the diffusion phenomena as mentioned previously…EDS or WDS analysis is mandatory.

Reply: Thank you for the comment. We corrected it in the revised version of the paper. Lines 528-563.

Remark 11: Number the conclusions.

Reply: Thank you for the comment. We corrected it in the revised version of the paper.

With best regards,

Dr. Vadim Tynchenko

Reviewer 2 Report

Errors and confusions

  1. the number of Authors is quite a lot (9 to be specific)
  2. spelling errors, (line 34, 107,
  3. on material and methods section the sentence containing “the customer formulated…” is confusing.
  4. On testing methods section: as “the heating rate of the powder mixture & the sintering temperature of the material” being specified as some of research outputs, the reader was unable to find any relevant discussion related to powder heating or sintering temperature throughout the paper.
  5. Redundancy of ideas and sentences is observed on the report, for instance the paragraph written at line 417-421 & line 467-470 are identical.
  6. line 482-484, “The results showed that in the NbStrip-1+VT-14 connection zone, a zone characterized by an 483 altered structure (a diffusion zone) with a width of 0.011 mm was found (Figure 10).” The reader observed a contradictory statement here.( figure 10 was labeled Connection area of 12KH18N10T stainless steel and copper)
  7. line 484-485, “Moreover, a diffusion zone was revealed at the boundary of the 12KH18N10T+M1 compound with a width of up to 0.005 mm (Figure 12)”. The reader observed a contradictory statement here. (figure 12 was labeled as VT-14+Nb connection microsection)

Comments

  1. lots of vague and fragmented sentences were observed throughout the paper, it is advisable to revise and rewrite some of sentences. For instance, lines 28,29,35, 135-137, 201,208,229,261,341-342, 364-365
  2. Some graphical representations are not clear to observe, it is advisable to use high resolution picture showing the necessary features. (figure 5, figure7, figure 8,). It was impossible to identify a copper interlayer on Figure 7 at the fracture, as it was claimed to be observed on niobium or steel surface.
  3. As the microstructure of titanium being specified in the paper as one of the judging criteria for acceptable weld, deep discussion on the microstructures obtained is expected. Authors are also advised to clearly show, if the obtained microstructure is complied with the standard specified in the paper (OST 97 9465-81).
  4. As Ultrasonic inspection being specified as a validation method for quality of the weld, the reader expects certain discussion on the results obtained while testing the specimens.
  5. As specified on Figure 3, Standard sample No. 4 from Interstate Standard 1497-84 (10 mm in diameter and 90 mm long) was the specimen standard promised to be employed throughout the tests, which was found to be contradicting with the test specimen presented and applied on real investigation (Figure 6 with The diameter of the test section was 6 mm, the length was 17 mm).
  6. The mechanical strength of Cu foil was specified to be 227MPa, which is quite small compared to the strength level reported to be achieved from the dissimilar joint. The reader expects logical and experimental explanation for the resulted improvement on tensile strength of Cu interlayer. (elemental diffusion is expected and needs to be experimentally verified. EDS might be appropriate here)
  7. It was concluded that copper was found on the fractured sites, experimental verification is advised to confirm the chemical composition on fractured sites.
  8. line 441-443 “The presence of diffusion zones at the steel + copper and copper + niobium interface was confirmed by the tensile tests results, which showed that the destruction of the samples in all cases occurred over the entire area of the samples over the less durable material, i.e., copper.” It is quite confusing to grasp the logic here as this statement presents that the existence of diffusion layer could be reveled using tensile testing results. Elemental analysis using EDS or any other method is advised here. 
  9. it is advisable to show stress strain curves for tensile tests performed on major specimens.
  10. line 473-475 “In addition, it was found that the microstructure of the microsection titanium component consisted of a mixture of (a+b)-phases and corresponded to type 4-6 of the 9 typical scales for (a+b)-titanium alloy microstructures, according to OST 97 9465-81.” Further explanation on the existence of α-β Ti phases corresponding to the microstructure presented by the authors is advised here.

Author Response

Dear Reviewer 2,

Thank you for your positive comments on our paper. You provided remarks about it, so we’ve made changes in the paper.

Remark 1: the number of Authors is quite a lot (9 to be specific).

Reply: Thank you for the comment. We would like to clarify that this work has been carried out over the past five years and only now the time has come to formalize the results. Throughout this time, each of the co-authors was involved in the performance of certain types of work. And, in our opinion, it would be wrong to exclude any of them from co-authorship. More detailed information about the types of work performed by each of the co-authors is presented in the “Author Contributions” section of the Meta information about the article in the system.

Remark 2: spelling errors, (line 34, 107).

Reply: Thank you for the comment. In line 34 presented welding modes. Line 107 shows the pressure conditions at which it is impossible to achieve sufficient welding forces for plastic deformation. We used MDPI English Editing Services (ID: english-31584).

Remark 3: on material and methods section the sentence containing “the customer formulated…” is confusing.

Reply: Thank you for the comment. We reformulated the phrase and tried to explain it. Lines 42-61, 84-90, 186-188.

Remark 4: On testing methods section: as “the heating rate of the powder mixture & the sintering temperature of the material” being specified as some of research outputs, the reader was unable to find any relevant discussion related to powder heating or sintering temperature throughout the paper.

Reply: Thank you for the comment. We changed it (line 307-310).

Remark 5: Redundancy of ideas and sentences is observed on the report, for instance the paragraph written at line 417-421 & line 467-470 are identical.

Reply: Thank you for the comment. We deleted line 467-470.

Remark 6: line 482-484, “The results showed that in the NbStrip-1+VT-14 connection zone, a zone characterized by an 483 altered structure (a diffusion zone) with a width of 0.011 mm was found (Figure 10).” The reader observed a contradictory statement here. ( figure 10 was labeled Connection area of 12KH18N10T stainless steel and copper).

Reply: Thank you for the comment. We added corrected link to Figure 15.

Remark 7: line 484-485, “Moreover, a diffusion zone was revealed at the boundary of the 12KH18N10T+M1 compound with a width of up to 0.005 mm (Figure 12)”. The reader observed a contradictory statement here. (figure 12 was labeled as VT-14+Nb connection microsection).

Reply: Thank you for the comment. We added corrected link to Figure 13.

Remark 8: lots of vague and fragmented sentences were observed throughout the paper, it is advisable to revise and rewrite some of sentences. For instance, lines 28,29,35, 135-137, 201,208,229,261,341-342, 364-365.

Reply: Thank you for the comment. We used MDPI English Editing Services (ID: english-31584).

Remark 9: Some graphical representations are not clear to observe, it is advisable to use high resolution picture showing the necessary features. (figure 5, figure7, figure 8,). It was impossible to identify a copper interlayer on Figure 7 at the fracture, as it was claimed to be observed on niobium or steel surface.

Reply: Thank you for the comment. We tried to get a better quality to figures. Lines 277-283.

Remark 10: As the microstructure of titanium being specified in the paper as one of the judging criteria for acceptable weld, deep discussion on the microstructures obtained is expected. Authors are also advised to clearly show, if the obtained microstructure is complied with the standard specified in the paper (OST 97 9465-81).

Reply: Thank you for the comment. In the paper, we did not assert that the change in the microstructure of titanium alloys during the welding process is considered as a criterion for the quality of weld welding. We stated that the structure of the welded samples had according to the atlas of structures of titanium alloys consists of a and b phases according to OST 97 9465-81.

Remark 11: As Ultrasonic inspection being specified as a validation method for quality of the weld, the reader expects certain discussion on the results obtained while testing the specimens.

Reply: Thank you for the comment. We corrected it in the revised version of the paper. Lines 349-361.

Remark 12: As specified on Figure 3, Standard sample No. 4 from Interstate Standard 1497-84 (10 mm in diameter and 90 mm long) was the specimen standard promised to be employed throughout the tests, which was found to be contradicting with the test specimen presented and applied on real investigation (Figure 6 with The diameter of the test section was 6 mm, the length was 17 mm).

Reply: Thank you for the comment. According to Figure 4 for standard samples No. 4. On their basis, strength tests were carried out to develop welding modes. And already from the sample Figure 7, for carrying out cyclic tests, cylindrical samples of solid section were made with the plane of connection of the layers normal to the axis of the sample. The diameter of the working part was 6 mm, the length was 17 mm, the roughness Ra was 0.62 μm.

Remark 13: The mechanical strength of Cu foil was specified to be 227MPa, which is quite small compared to the strength level reported to be achieved from the dissimilar joint. The reader expects logical and experimental explanation for the resulted improvement on tensile strength of Cu interlayer. (elemental diffusion is expected and needs to be experimentally verified. EDS might be appropriate here).

Reply: Thank you for the comment. We added the explanations in the revised version of the paper. Lines 254-271.

Remark 14: It was concluded that copper was found on the fractured sites, experimental verification is advised to confirm the chemical composition on fractured sites.

Reply: Thank you for the comment. Determination of copper on broken samples was carried out visually. After the rupture of the samples, Figure 10. The copper coating from the foil is visible on both contacting sides of the materials to be welded.

Remark 15: line 441-443 “The presence of diffusion zones at the steel + copper and copper + niobium interface was confirmed by the tensile tests results, which showed that the destruction of the samples in all cases occurred over the entire area of the samples over the less durable material, i.e., copper.” It is quite confusing to grasp the logic here as this statement presents that the existence of diffusion layer could be reveled using tensile testing results. Elemental analysis using EDS or any other method is advised here.

Reply: Thank you for the comment. We added the explanations in the revised version of the paper. Lines 412-413, 528-563.

Remark 16: it is advisable to show stress strain curves for tensile tests performed on major specimens.

Reply: Thank you for the comment. When working out the welding modes, we determined the tensile strength (σ) and did not use the recording of rupture curves.

Remark 17: line 473-475 “In addition, it was found that the microstructure of the microsection titanium component consisted of a mixture of (a+b)-phases and corresponded to type 4-6 of the 9 typical scales for (a+b)-titanium alloy microstructures, according to OST 97 9465-81.” Further explanation on the existence of α-β Ti phases corresponding to the microstructure presented by the authors is advised here.

Reply: Thank you for the comment. We did not mention in the article that the change in the microstructure of titanium alloys during the welding process was considered as a criterion for the quality of seam welding. We stated that the structure of already welded samples had, according to the atlas of structures of titanium alloys, consists of a and b phases.

With best regards,

Dr. Vadim Tynchenko

Reviewer 3 Report

The work of this paper is of interest, but the manuscript needs some work for publication.

  1. Please check Fig 7 and mark the axes labels and transform x-axis to y-axis.
  2. Please transform x-axis to y-axis in Fig 7 and 8.

3.The quality of the figure 9 to 17 need to improve.

  1. Please give the deeply description of mechanism for the Cu-Nb and Cu-12KH18N10T interface fracture under load..

5.The conclusion need complete revision. Some of the main results in the discussion section could be included in the conclusions.

Author Response

Dear Reviewer 3,

Thank you for your positive comments on our paper. You provided remarks about it, so we’ve made changes in the paper.

Remark 1: Please check Fig 7 and mark the axes labels and transform x-axis to y-axis.

Reply: Thank you for the comment. We corrected it. Line 417.

Remark 2: Please transform x-axis to y-axis in Fig 7 and 8.

Reply: Thank you for the comment. We corrected it. Lines 417, 426.

Remark 3: The quality of the figure 9 to 17 need to improve.

Reply: Thank you for the comment. The initial research was done several years ago and we have no way of obtaining new images. However, we have tried to improve the quality of the figures.

Remark 4: Please give the deeply description of mechanism for the Cu-Nb and Cu-12KH18N10T interface fracture under load.

Reply: Thank you for the comment. Lines 412-413, 467-469.

Remark 5: The conclusion need complete revision. Some of the main results in the discussion section could be included in the conclusions.

Reply: Thank you for the comment. We added a minor change in conclusion but tried to rewrite it.

With best regards,

Dr. Vadim Tynchenko

Reviewer 4 Report

1 – Figure 1, 2, 3, and 6. I would recommend the authors to label the materials in these figures. This should help the readers to understand the geometry of the samples described in the context.

2 – The authors identify the “diffused layer” solely by their distinctive contrast and morphology in the images. However, the depth of diffusion can be measured by many widely available techniques such as EDS or EPMA. I recommend the authors to add some compositional analyses.

3 – Please insert an X-axis title to Figure 7.

4 – Figure 9, 10, 13, 14, 16 and 17. The contrast of these figures should be adjusted so that each phase reveals their structure. I think that should help the readers greatly to understand the purpose of these figures as well. I encourage the authors to replace these images for a better quality.

5 – Selection of the element for the interlayer must be based on the understanding of intermetallic phases and diffusion kinetics of the used elements. This paper lacks the scientific discussion of why the elements are chosen and what other elements can be used for better(or worse) performance.

Author Response

Dear Reviewer 4,

Thank you for your positive comments on our paper. You provided remarks about it, so we’ve made changes in the paper.

Remark 1: Figure 1, 2, 3, and 6. I would recommend the authors to label the materials in these figures. This should help the readers to understand the geometry of the samples described in the context.

Reply: Thank you for the comment. We corrected it. Lines 277-348.

Remark 2: The authors identify the “diffused layer” solely by their distinctive contrast and morphology in the images. However, the depth of diffusion can be measured by many widely available techniques such as EDS or EPMA. I recommend the authors to add some compositional analyses.

Reply: Thank you for the comment. We added explanations. Lines 528-563.

Remark 3: Please insert an X-axis title to Figure 7.

Reply: Thank you for the comment. We corrected it (line 417, 426).

Remark 4: Figure 9, 10, 13, 14, 16 and 17. The contrast of these figures should be adjusted so that each phase reveals their structure. I think that should help the readers greatly to understand the purpose of these figures as well. I encourage the authors to replace these images for a better quality.

Reply: Thank you for the comment. The initial research was done several years ago and we have no way of obtaining new images. However, we have tried to improve the quality of the figures.

Remark 5: Selection of the element for the interlayer must be based on the understanding of intermetallic phases and diffusion kinetics of the used elements. This paper lacks the scientific discussion of why the elements are chosen and what other elements can be used for better (or worse) performance.

Reply: Thank you for the comment. We added explanations about it (lines 160-176).

With best regards,

Dr. Vadim Tynchenko

Round 2

Reviewer 1 Report

The authors have clearly made changes and/or modifications to the manuscript.

It is always important to show respect to the review process and to incorporate a letter (reply to the reviewers) indicating all changes made to the original manuscript; additionally, to point out the suggested corrections and/or modifications that were not possible to carry out!

Reviewer 2 Report

Weld the workpieces in the following modes:  Vacuum degree is about 1.3 Pa, this degree is much lower, the titanium alloy is very easy react with oxygen under this atmosphere. Does the titanium react with the oxygen in your work?  
In table 12, the results of the cycle is confused, we do not know which sample is direct brazing or with  intermediate layers? 
The microstructure of the brazed joint and their phase constitutes is not clear.

The main problem of this submission,  no comparision and detailed analysis on the welding joints which including the interlayer.

Dear Reviewer 2,

Thank you for your comments on our paper. You provided remarks about it, so we’ve prepared answers.

Remark 1:  Weld the workpieces in the following modes:  Vacuum degree is about 1.3 Pa, this degree is much lower, the titanium alloy is very easy react with oxygen under this atmosphere. Does the titanium react with the oxygen in your work?

Reply: Thank you for the comment. The diffusion welding method we use involves welding in full vacuum. In the process of welding, the pressure in the chamber is 18MPa. Thus, the titanium does not react with the oxygen in the process of diffusion welding, since it is in the chamber at full vacuum. Unfortunately, at the moment we cannot check the mode you proposed, because it is very labor intensive. Also, in this work was revealed that the oxygen content in the titanium component during diffusion welding of the bimetal does not change according to the selected modes. Moreover, the bimetal samples, presented in terms of oxygen content, meet the technical requirements.

Remark 2:  In table 12, the results of the cycle is confused, we do not know which sample is direct brazing or with intermediate layers?

Reply: Thank you for the comment. All samples shown in Table 12 have intermediate layers (Figure 7). The results of the mechanical tests of samples (the bimetallic VT-14 + Nb + Cu + 12KH18N10T compound) under static and cyclic loading are shown in Table 12.

Remark 3:  The microstructure of the brazed joint and their phase constitutes is not clear.

Reply: Thank you for the comment. The micro and macro structures of the weld are shown in Figures 12-15. Microsection of welded solid-phase VT-14 + Nb + Cu + 12KH18N10T joint is shown in Figure 12. In this work, the technological process is being worked out with further mechanical testing, therefore the main characteristic is strength. For a detailed answer to the question "their phase constitutes", a number of special separate studies are required, that are not in the scope of the presented paper .

Remark 4:   The main problem of this submission, no comparision and detailed analysis on the welding joints which including the interlayer.

Reply: Thank you for the comment. The main topic of the presented article is to obtain an equal-strength joint VT-14 and 12KH18N10T using niobium and copper interlayers, as well as to study the strength of the resulting joint. Further studies carried out by the authors confirmed the assumptions made, in particular, the passage of diffusion processes. The authors did not present complete description of the joint zone and interlayer zones studies in this work. However, in the future we will take this remark into account, and in subsequent works we will add the appropriate section.

With best regards,

Dr. Vadim Tynchenko